# Personalized Federated Learning via Variational Message Passing

## Abstract

Conventional federated learning (FL) aims to train a unified machine learning model that fits data distributed across various agents. However, statistical heterogeneity arising from diverse data resources renders the single global model trained by FL ineffective for all clients. Personalized federated learning (pFL) has been proposed to primarily address this challenge by tailoring individualized models to each client's specific dataset while integrating global information during feature aggregation. Achieving efficient pFL necessitates the accurate estimation of global feature information across all the training data. Nonetheless, balancing the personalization of individual models with the global consensus of feature information remains a significant challenge in existing approaches. In this paper, we propose *pFedVMP*, a novel pFL approach that employs variational message passing (VMP) to design feature aggregation protocols. By leveraging the mean and covariance, *pFedVMP* yields more precise estimates of the distributions of model parameters and global feature centroids. Additionally, pFedVMP is effective in boosting training accuracy and preventing overfitting by regularizing local training with global feature centroids. Extensive experiments on heterogeneous data conditions demonstrate that *pFedVMP* surpasses state-of-the-art methods in both effectiveness and fairness.

## 1 Introduction

Federated learning (FL) is a promising distributed learning paradigm that enables clients to collaboratively train models without uploading private data, thereby protecting local data privacy (McMahan et al., 2017). In the standard FL framework, clients train a uniform learning model using local datasets and employ linear model aggregation to combine these local models, assuming that while the local data across clients may differ in size, they generally share similar underlying distributions. This assumption potentially leads to a global model that performs reasonably well when deployed on each client. However, in practice, local data distributions vary due to diverse sources and data quality, resulting in a phenomenon known as statistical heterogeneity of training data (Zhao et al., 2018). This heterogeneity makes the globally optimal model perform poorly on local datasets.

Personalized federated learning (pFL) has been introduced to address the challenge of statistical heterogeneity by training personalized models that better align with each client's local dataset, rather than relying on a single global model. This is accomplished through an iterative process that alternates between two key steps: (1) Aggregating shared *feature* information from local models to capture the underlying patterns present across local datasets, and (2) Developing tailored models for clients to meet their specific objectives by leveraging the aggregated global information. Existing work often concentrates exclusively on either personalized feature information (e.g., FedPer (Arivazhagan et al., 2019), FedPep (Collins et al., 2021)) or global feature aggregation (e.g., FedROD (Chen & Chao, 2022)). This leads to neglect of balancing personalization and global consistency. To address this issue, several pFL approaches incorporate global information to improve local feature extraction. For example, FedProto (Tan et al., 2022) and FedPAC (Xu et al., 2023) align local feature representations closely with their respective centroids, where the global centroids are estimated by averaging the feature samples. However, due to statistical heterogeneity of training data, the arithmetic mean of feature samples deviates from the ground-truth centroids (Al-Shedivat et al., 2021; Guo et al., 2023), which in turn might degrade the accuracy of local feature extraction. To tackle this challenge, Bayesian estimation methods have been adopted in FL. For example, FedPA

(Al-Shedivat et al., 2021) and FedEP (Guo et al., 2023) design model aggregation protocols based on Bayesian principles. By leveraging the mean and covariance of model parameters, these methods achieve more accurate estimate of the global model.

In this paper, we propose a pFL approach, termed *pFedVMP*, which leverages a variational message passing approach for feature aggregation. This method conceptualizes both model parameters and feature centroids as random variables and aggregates their distributions via a maximum-a-posteriori (MAP) criterion to update the global model. To simplify the MAP estimation, we utilize variational inference to decompose the joint density distribution of the variables using multiplicative factors. By leveraging the mean and covariance, the variational message passing rules yield more precise estimates of the distributions of model parameters and global feature centroids. Furthermore, the variational message passing algorithm yields a model update rule that aligns with a regularized local optimization framework, utilizing global feature centroids to enhance personalized model training. This approach is validated as effective in improving training accuracy and preventing overfitting. The **key contributions** are summarized as follows:

- We develop a unified probabilistic framework that integrates both model parameters and feature centroids, proposing a pFL approach based on variational message passing, termed *pFedVMP*, to address statistical data heterogeneity.

- *pFedVMP* provides more precise estimates of the distributions of model parameters and global feature centroids by utilizing the means and covariances. This approach achieves a balance between global feature estimation and local model personalization in pFL.

- We perform extensive experiments under various data heterogeneity settings. The results demonstrate that *pFedVMP* outperforms state-of-the-art methods in terms of both effectiveness and fairness.

## 2 RELATED WORK

**FL under statistical heterogeneity of data.** The FL framework was initially proposed by McMahan et al. (2017). Subsequent studies, such as those by (Khaled et al., 2020; Zhao et al., 2018), have underscored the significant impact of statistical heterogeneity in training data on the convergence rate and learning accuracy of FL models. This challenge has continuously drawn attention in the research community. Various strategies have been proposed to address this issue, including regularized local training using global information (Li et al., 2020; Durmus et al., 2021; Li et al., 2021a), local bias correction (Karimireddy et al., 2020), data augmentation (Li et al., 2022; Yoon et al., 2021), and knowledge distillation (Zhu et al., 2021; Lin et al., 2020).

The drive to address statistical heterogeneity has significantly shaped the development of pFL approaches, which train localized models tailored to diverse local data distributions (Dai et al., 2023; Zhang et al., 2023a; Islam et al., 2024; Hanzely & Richtárik, 2020). Initial pFL strategies typically involved a straightforward extension of linear model aggregation similar to conventional FL (Deng et al., 2020; Hanzely & Richtárik, 2020). Since then, more sophisticated pFL protocols have emerged, drawing inspiration from advanced learning mechanisms, such as meta-learning (Fallah et al., 2020; Chen et al., 2018), multi-task learning (Smith et al., 2017; T Dinh et al., 2020; Li et al., 2021b), and model splitting strategies (Arivazhagan et al., 2019; Collins et al., 2021; Chen & Chao, 2022; Liang et al., 2020; Oh et al., 2022; Zhang et al., 2023b). While these approaches have improved the performance on the heterogeneous data, they may still be prone to overfitting, particularly when the training dataset size is small (Zhang et al., 2023d;a).

**Federated Representation Learning.** Several pFL approaches, such as FedSR(Nguyen et al., 2022), FedCiR(Li et al., 2024), FedProto(Tan et al., 2022), MOON(Li et al., 2021a), FedCP(Zhang et al., 2023c), FedPAC(Xu et al., 2023), and GPFL(Zhang et al., 2023a), integrated representation learning by learning a client-invariant representation. This representation maintains a consistent conditional distribution across clients and is leveraged in local training as a foundation model, which is shown effective in preventing overfitting. Specifically, FedSR and FedCiR computed global feature distributions by using probabilistic networks and generative networks, respectively, which are not directly applicable to pFL. In contrast, FedProto and FedPAC estimated the mean of global feature distributions by averaging local feature samples. MOON aligns the local and global representations by maximizing their similarity. GPFL embedded features in a representation space and subsequently

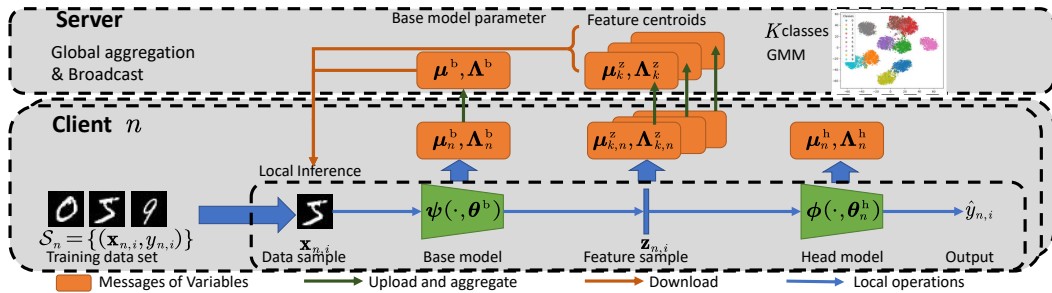

Figure 1: Schematic view of pFedVMP.

estimated global feature distributions implicitly with the embedding dictionary. In contrast, pFed-VMP leverages the covariance estimates of feature representations in aggregation, subsequently leading to a more robust estimation of the base model.

**Bayesian Federated Learning.** Bayesian federated learning (BFL) was proposed to improve the robustness and learning performance, particularly on small-scale datasets (Cao et al., 2023). BFL can be broadly categorized into client-side BFL and server-side BFL based on federated learning architectures. Client-side BFL focuses on learning Bayesian local models on client nodes, including BNFed (Yurochkin et al., 2019), pFedGP (Achituve et al., 2021), and pFedBayes (Zhang et al., 2023d). Specifically, BNFed and pFedGP train Bayesian nonparametric models, while pFedBayes trains Bayesian neural networks. In contrast, server-side BFL aggregates local updates for global models using Bayesian methods, including FedPA (Al-Shedivat et al., 2021), FedEP (Guo et al., 2023), QLSD (Vono et al., 2022), pFedBreD (Shi et al., 2024). This branch of methods formulates model training as model inference tasks and computes the maximum-a-posterior (MAP) estimator (Al-Shedivat et al., 2021; Guo et al., 2023; Vono et al., 2022). In the FL setups, the distributed nature of datasets among clients prevents direct computation of model posterior distributions. FedPA approximated the posterior distribution into the product of distributions with respect to local datasets during local model training. FedEP developed the Bayesian model aggregation rule by using expectation propagation. QLSD extended the approach in FedPA with the quantized Langevin stochastic dynamics for local update. pFedBreD incorporates personalized prior knowledge for meta-learning. However, the above BFL methods do not utilize global feature centroids to guide local model training, which limits their ability to effectively address data heterogeneity. In contrast, pFedVMP considers both model parameters and feature centroids, guiding local training through a regularization term based on global feature centroids, thereby enhancing learning performance.

## 3 SYSTEM MODEL AND PROBLEM FORMULATION

We consider an FL system to train a supervised classification model under the coordination of a parameter server (PS) and $N$ clients. Each client $n$ owns its local dataset $\mathcal{S}_n$ with $|\mathcal{S}_n| = S_n$ labeled data points. The $i$-th data point in $\mathcal{S}_n$ is denoted by $(\mathbf{x}_{n,i}, y_{n,i})$, where $\mathbf{x}_{n,i}$ denotes the data sample, and $y_{n,i} \in \{1, \cdots, K\}$ denotes the label of $\mathbf{x}_{n,i}$. Let $\mathcal{S} = \bigcup_{n=1}^{N} \mathcal{S}_n$ denote the collection of the training data from all the clients, which is assumed to be categorized into $K$ classes and the data in each class is independent identically distributed (i.i.d.) from an unknown distribution. We denote the overall data distribution as a mixture distribution $p_{\mathcal{D}}(\mathbf{x}, y)$. We assume that the data size of each label class at each client is known at the PS beforehand.

In practice, data heterogeneity across clients results in heterogeneous statistics of local data, including their means, variances, etc. This discrepancy leads to distinct marginal distributions for local datasets, presenting a challenge known as the statistical heterogeneity of training data (Zhao et al., 2018; Arivazhagan et al., 2019; Tan et al., 2022). Such heterogeneity invalidates the common i.i.d. data assumption in the machine learning literature, arising challenges in model bias and overfitting.

In this work, we employ pFL to address the challenge of statistical heterogeneity. Instead of training a uniform global model that tries to fit all the local datasets, pFL aims to train personalized models tailored to each client's individual dataset. As shown in Fig. 1, the clients share a common base model to extract global feature representations and learn a personalized head model to enhance

performance on their local datasets. Specifically, on any client $n$, its local network can be divided into two parts: 1) a *base* model $\psi$ parameterized by $\boldsymbol{\theta}^{\mathrm{b}}$ to extract the feature $\mathbf{z}_{n,i}$ corresponding to the input data sample $\mathbf{x}_{n,i}$, given by $\mathbf{z}_{n,i} = \boldsymbol{\psi}(\mathbf{x}_{n,i}, \boldsymbol{\theta}^{\mathrm{b}})$; 2) a *head* model $\phi$ parameterized by $\boldsymbol{\theta}_n^{\mathrm{h}}$ to map the feature $\mathbf{z}_{n,i}$ to the label $\hat{y}_{n,i}$, given by $\hat{y}_{n,i} = \phi(\mathbf{z}_{n,i}, \boldsymbol{\theta}_n^{\mathrm{h}})$. Given a base model specified by the parameter $\boldsymbol{\theta}^{\mathrm{b}}$, the collection of feature samples with respect to (w.r.t.) the $n$-th training dataset $\mathcal{S}_n$ is denoted by $\mathcal{Z}_n = \{(\mathbf{z}_{n,i}, y_{n,i}); i = 1, \ldots, S_n\}$, where $\mathbf{z}_{n,i} = \boldsymbol{\psi}(\mathbf{x}_{n,i}, \boldsymbol{\theta}^{\mathrm{b}})$ is the $i$-th feature sample on client $n$, and the local dataset $\mathcal{S}_n$. As the training data encompasses $K$ classes, we can categorize the corresponding features based on the class of the input data, represented as $\mathcal{Z}_n = \bigcup_{k=1}^K \mathcal{Z}_{k,n}$, where $\mathcal{Z}_{k,n} = \{(\mathbf{z}_{n,i}, k)\}$ denotes the set of feature samples corresponding to class $k$. Let $Z_n$ and $Z_{k,n}$ denote the total number of features in $\mathcal{Z}_n$ and the number of features in each class subset $\mathcal{Z}_{k,n}$, respectively. Let $\mathbf{z}_k$ denote the global centroid of the features of class $k$, and $\mathbf{z}_{k,n}$ denote the local centroid of the features of class $k$ on client $n$. Due to the heterogeneous and non-shareable nature of local data in the FL setting, the local base model tends to overfit the local data, causing the local feature centroid $\mathbf{z}_{k,n}$ to diverge from the global feature centroid $\mathbf{z}_k$ and resulting in poor performance on subsequent classification tasks.

Before introducing the proposed approach, we formulate the distributed optimization problem for the pFL system. Following the Bayesian FL problem formulation (Al-Shedivat et al., 2021; Guo et al., 2023), we model the parameters $\boldsymbol{\theta}^{\mathrm{b}}, \{\boldsymbol{\theta}_n^{\mathrm{h}}\}$ and the global feature centroids $\{\mathbf{z}_k\}$ as random variables. Our goal is to solve a maximum *a posteriori* probability (MAP) estimation problem w.r.t. the variables $(\boldsymbol{\theta}^{\mathrm{b}}, \{\boldsymbol{\theta}_n^{\mathrm{h}}\}, \{\mathbf{z}_k\})$, given by

$$\max_{\boldsymbol{\theta}^{\mathrm{b}}, \{\boldsymbol{\theta}_n^{\mathrm{h}}\}, \{\mathbf{z}_k\}} p(\boldsymbol{\theta}^{\mathrm{b}}, \{\boldsymbol{\theta}_n^{\mathrm{h}}\}, \{\mathbf{z}_k\} | \mathcal{S}). \tag{1}$$

In general, performing exact inference on the distribution $p(\boldsymbol{\theta}^{\mathrm{b}}, \{\boldsymbol{\theta}_n^{\mathrm{h}}\}, \{\mathbf{z}_k\} | \mathcal{S})$ is intractable due to the high dimensionality of the variables and the unshared nature of the local datasets.

# 4 PROPOSED FRAMEWORK

In the following sections, we introduce approximate inference to simplify the optimization process and propose a new approach, termed personalized Federated Learning via Variational Massage Passing (pFedVMP), for efficient feature aggregation. Motivated by variational inference (Minka, 2001), we use a decomposable surrogate distribution $q(\boldsymbol{\theta}^{\mathrm{b}}, \{\boldsymbol{\theta}_n^{\mathrm{h}}\}, \{\mathbf{z}_k\})$ to approximate the distribution $p$. Specifically, we convert the original problem in eq. (1) as follows:

$$(\text{P1}) \min_{q(\boldsymbol{\theta}^{\mathrm{b}}, \{\boldsymbol{\theta}_n^{\mathrm{h}}\}, \{\mathbf{z}_k\})} D_{\mathrm{KL}}(p(\boldsymbol{\theta}^{\mathrm{b}}, \{\boldsymbol{\theta}_n^{\mathrm{h}}\}, \{\mathbf{z}_k\} | \mathcal{S}) \| q(\boldsymbol{\theta}^{\mathrm{b}}, \{\boldsymbol{\theta}_n^{\mathrm{h}}\}, \{\mathbf{z}_k\})), \tag{2}$$

where $D_{\mathrm{KL}}(\cdot \| \cdot)$ denotes the KL-divergence. The chosen surrogate distribution $q(\boldsymbol{\theta}^{\mathrm{b}}, \{\boldsymbol{\theta}_n^{\mathrm{h}}\}, \{\mathbf{z}_k\})$ is required to admit a decomposable form as:

$$q(\boldsymbol{\theta}^{\mathrm{b}}, \{\boldsymbol{\theta}_n^{\mathrm{h}}\}, \{\mathbf{z}_k\}) \propto q(\boldsymbol{\theta}^{\mathrm{b}}) q(\{\boldsymbol{\theta}_n^{\mathrm{h}}\}) q(\{\mathbf{z}_k\}), \tag{3}$$

where $q(\boldsymbol{\theta}^{\mathrm{b}}), q(\{\boldsymbol{\theta}_n^{\mathrm{h}}\}), q(\{\mathbf{z}_k\})$ denote the global factors for the **base** parameters $\boldsymbol{\theta}^{\mathrm{b}}$, the **head** parameters $\boldsymbol{\theta}_n^{\mathrm{h}}$, and the **feature centroids** $\{\mathbf{z}_k\}$, respectively. These marginal distribution can be further factorized as the products of prior and local likelihood distributions as

$$q(\boldsymbol{\theta}^{\mathrm{b}}) \propto q_{\mathrm{pri}}(\boldsymbol{\theta}^{\mathrm{b}}) \prod_{n=1}^N q_n(\boldsymbol{\theta}^{\mathrm{b}}), \; q(\{\boldsymbol{\theta}_n^{\mathrm{h}}\}) \propto \prod_{n=1}^N q_{\mathrm{pri}}(\boldsymbol{\theta}_n^{\mathrm{h}}) q_n(\boldsymbol{\theta}_n^{\mathrm{h}}), \; q(\{\mathbf{z}_k\}) \propto q_{\mathrm{pri}}(\{\mathbf{z}_k\}) \prod_{n=1}^N q_n(\{\mathbf{z}_k\}), \tag{4}$$

where $q_{\mathrm{pri}}(\boldsymbol{\theta}^{\mathrm{b}}), q_{\mathrm{pri}}(\boldsymbol{\theta}_n^{\mathrm{h}}), q_{\mathrm{pri}}(\{\mathbf{z}_k\})$ denote the prior factors for $\boldsymbol{\theta}^{\mathrm{b}}, \boldsymbol{\theta}_n^{\mathrm{h}}$, and $\{\mathbf{z}_k\}$, respectively; and $q_n(\boldsymbol{\theta}^{\mathrm{b}}), q_n(\boldsymbol{\theta}_n^{\mathrm{h}}), q_n(\{\mathbf{z}_k\})$ denote the local likelihood factors with given the local dataset $\mathcal{S}_n$ on client $n$ for $\boldsymbol{\theta}^{\mathrm{b}}, \boldsymbol{\theta}_n^{\mathrm{h}}$, and $\{\mathbf{z}_k\}$, respectively.

As shown in Fig. 1, in each training iteration the client share the local information on the base parameters $\boldsymbol{\theta}^{\mathrm{b}}$ and the feature centroids $\{\mathbf{z}_k\}$ to the server for aggregation, while keeping the head parameters $\{\boldsymbol{\theta}_n^{\mathrm{h}}\}$ local. Specifically, the clients and the PS update the factors in eq. (3) and eq. (4) corporately to find the optimal $(\boldsymbol{\theta}^{\mathrm{b}}, \{\boldsymbol{\theta}_n^{\mathrm{h}}\}, \{\mathbf{z}_k\})$ that maximize the objective in (P1). In the following, we shall detail the concrete updating expressions for specific choices of the distributions.

### 4.1 VARIATIONAL INFERENCE

We first discuss the factors for the model parameters $\boldsymbol{\theta}^{\mathrm{b}}, \boldsymbol{\theta}_n^{\mathrm{h}}$. Following previous works on variational inference (Minka, 2001; Al-Shedivat et al., 2021; Guo et al., 2023), we use the multivariate Gaussian distribution as the variational family for the factors w.r.t. $\boldsymbol{\theta}^{\mathrm{b}}, \boldsymbol{\theta}_n^{\mathrm{h}}$, given by $q_{\mathrm{pri}}(\boldsymbol{\theta}^{\mathrm{b}}) = \mathcal{N}(\boldsymbol{\mu}_{\mathrm{pri}}^{\mathrm{b}}, (\boldsymbol{\Lambda}_{\mathrm{pri}}^{\mathrm{b}})^{-1})$, $q_n(\boldsymbol{\theta}^{\mathrm{b}}) = \mathcal{N}(\boldsymbol{\mu}_n^{\mathrm{b}}, (\boldsymbol{\Lambda}_n^{\mathrm{b}})^{-1})$, $q_{\mathrm{pri}}(\boldsymbol{\theta}_n^{\mathrm{h}}) = \mathcal{N}(\boldsymbol{\mu}_{\mathrm{pri}}^{\mathrm{h}}, (\boldsymbol{\Lambda}_{\mathrm{pri}}^{\mathrm{h}})^{-1})$, $q_n(\boldsymbol{\theta}_n^{\mathrm{h}}) = \mathcal{N}(\boldsymbol{\mu}_n^{\mathrm{h}}, (\boldsymbol{\Lambda}_n^{\mathrm{h}})^{-1})$, where $(\boldsymbol{\mu}_{\mathrm{pri}}^{\mathrm{b}}, \boldsymbol{\Lambda}_{\mathrm{pri}}^{\mathrm{b}})$, $(\boldsymbol{\mu}_n^{\mathrm{b}}, \boldsymbol{\Lambda}_n^{\mathrm{b}})$, $(\boldsymbol{\mu}_{\mathrm{pri}}^{\mathrm{h}}, \boldsymbol{\Lambda}_{\mathrm{pri}}^{\mathrm{h}})$, $(\boldsymbol{\mu}_n^{\mathrm{h}}, \boldsymbol{\Lambda}_n^{\mathrm{h}})$ denote the mean vectors and the precision matrices of the factors $q_{\mathrm{pri}}(\boldsymbol{\theta}^{\mathrm{b}})$, $q_n(\boldsymbol{\theta}^{\mathrm{b}})$, $q_{\mathrm{pri}}(\boldsymbol{\theta}_n^{\mathrm{h}})$, $q_n(\boldsymbol{\theta}_n^{\mathrm{h}})$, respectively. We assume that the head parameters $\{\boldsymbol{\theta}_n^{\mathrm{h}}\}$ share the same prior distribution between the clients. Since the model parameters has high dimensions, we formulate the precision matrices $(\boldsymbol{\Lambda}_{\mathrm{pri}}^{\mathrm{b}}, \boldsymbol{\Lambda}_n^{\mathrm{b}}, \boldsymbol{\Lambda}_{\mathrm{pri}}^{\mathrm{h}}, \boldsymbol{\Lambda}_n^{\mathrm{h}})$ as diagonal matrices to reduce the computation complexity.

We now discuss the factors for the feature centroids $\{\mathbf{z}_k\}$. Following the works in representation learning (Yin et al., 2020), we use the Gaussian mixture (GM) distribution as the variational family for the factor related to the feature centroids $\{\mathbf{z}_k\}$. Specifically, for $q_n(\{\mathbf{z}_k\})$, we have

$$q_n(\{\mathbf{z}_k\}) = \sum_{k=1}^{K} q_n(\{\mathbf{z}_k\}, y_k) = \sum_{k=1}^{K} q_n(y_k) q_n(\mathbf{z}_k), \tag{5}$$

where $q_n(y)$ is the weight of the $k$-th component satisfying $\sum_{k=1}^{K} q_n(y_k) = 1$, representing the probability of the data belonging to class $k$ on client $n$, and $q_n(\mathbf{z}_k)$ is a multivariate Gaussian distribution, given by $q_n(\mathbf{z}_k) = \mathcal{N}(\boldsymbol{\mu}_{k,n}^{\mathrm{z}}, (\boldsymbol{\Lambda}_{k,n}^{\mathrm{z}})^{-1})$ with a mean $\boldsymbol{\mu}_{k,n}^{\mathrm{z}}$ and a precision matrix $\boldsymbol{\Lambda}_{k,n}^{\mathrm{z}}$. The prior distribution $q_{\mathrm{pri}}(\{\mathbf{z}_k\})$ is also set as a GM distribution, given by $q_{\mathrm{pri}}(\{\mathbf{z}_k\}) = \frac{1}{K} \sum_{k=1}^{K} q_{\mathrm{pri}}(\mathbf{z}_k)$, where the distribution of each component $q_{\mathrm{pri}}(\mathbf{z}_k)$ is a unit Gaussian distribution, i.e., $q_{\mathrm{pri}}(\mathbf{z}_k) = \mathcal{N}(\mathbf{0}, \mathbf{I})$. In eq. (4), we see that the global factor $q(\mathbf{z}_k)$ is a product of the local factors $q_n(\mathbf{z}_k)$ and the prior $q_{\mathrm{pri}}(\mathbf{z}_k)$, i.e., a product of $N + 1$ GM distributions, involving computing $K^{N+1}$ Gaussian components, leading an unbearable computation complexity. Thus, we turn to combine the components for each class $k$ separately, resulting in an aggregation of multiple Gaussian distributions for each class $k$. The details are discussed in Section 4.3.

### 4.2 LOCAL OPTIMIZATION PROBLEM

Based on the previous discussions on the factorization of the approximation distribution $q$, we are now ready to present the local optimization problem for each client. Let $q_n(\boldsymbol{\theta}^{\mathrm{b}}, \boldsymbol{\theta}_n^{\mathrm{h}}, \{\mathbf{z}_k\}) \propto q_n(\boldsymbol{\theta}^{\mathrm{b}}) q_n(\boldsymbol{\theta}_n^{\mathrm{h}}) q_n(\{\mathbf{z}_k\})$ denote the local factor for client $n$, and define the cavity factors of $\boldsymbol{\theta}^{\mathrm{b}}$, $\boldsymbol{\theta}_n^{\mathrm{h}}$, $\{\mathbf{z}_k\}$ as

$$q_{-n}(\boldsymbol{\theta}^{\mathrm{b}}) \propto \frac{q(\boldsymbol{\theta}^{\mathrm{b}})}{q_n(\boldsymbol{\theta}^{\mathrm{b}})}, q_{-n}(\{\boldsymbol{\theta}_n^{\mathrm{h}}\}) \propto \frac{q(\{\boldsymbol{\theta}_n^{\mathrm{h}}\})}{q_n(\boldsymbol{\theta}_n^{\mathrm{h}})}, q_{-n}(\{\mathbf{z}_k\}) \propto \frac{q(\{\mathbf{z}_k\})}{q_n(\{\mathbf{z}_k\})}. \tag{6}$$

We further express the distribution $q(\boldsymbol{\theta}^{\mathrm{b}}, \{\boldsymbol{\theta}_n^{\mathrm{h}}\}, \{\mathbf{z}_k\})$ as $q(\boldsymbol{\theta}^{\mathrm{b}}, \{\boldsymbol{\theta}_n^{\mathrm{h}}\}, \{\mathbf{z}_k\}) \propto q_n(\boldsymbol{\theta}^{\mathrm{b}}, \boldsymbol{\theta}_n^{\mathrm{h}}, \{\mathbf{z}_k\}) q_{-n}(\boldsymbol{\theta}^{\mathrm{b}}) q_{-n}(\{\mathbf{z}_k\}) q_{-n}(\boldsymbol{\theta}_n^{\mathrm{h}})$. On client $n$, by fixing the cavity factors $q_{-n}(\boldsymbol{\theta}^{\mathrm{b}})$, $q_{-n}(\{\boldsymbol{\theta}_n^{\mathrm{h}}\})$, $q_{-n}(\{\mathbf{z}_k\})$, we have the following local problem for client $n$:

$$\text{(P2)} \quad \min_{q_n(\boldsymbol{\theta}^{\mathrm{b}}, \boldsymbol{\theta}_n^{\mathrm{h}}, \{\mathbf{z}_k\})} D_{\mathrm{KL}}(p(\boldsymbol{\theta}^{\mathrm{b}}, \{\boldsymbol{\theta}_n^{\mathrm{h}}\}, \{\mathbf{z}_k\} | \mathcal{S}) \| q_n(\boldsymbol{\theta}^{\mathrm{b}}, \boldsymbol{\theta}_n^{\mathrm{h}}, \{\mathbf{z}_k\}) q_{-n}(\boldsymbol{\theta}^{\mathrm{b}}) q_{-n}(\{\mathbf{z}_k\}) q_{-n}(\{\boldsymbol{\theta}_n^{\mathrm{h}}\})), \tag{7}$$

where $p$ is the joint distribution defined in eq. (1). In general, with given the cavity distribution $q_{-n}$, client $n$ aims to find an optimal distribution $q_n$ to minimize the local objective in eq. (7). The PS then aggregates the updated factors $\{q_n\}$ and obtains the estimate of $(\boldsymbol{\theta}^{\mathrm{b}}, \boldsymbol{\theta}_n^{\mathrm{h}}, \{\mathbf{z}_k\})$ by solving (P1).

In practice, the statistical property of the local dataset $\mathcal{S}_n$ are different, leading to the issue of statistical heterogeneity. Statistical heterogeneity causes biased local estimation of $(\boldsymbol{\theta}^{\mathrm{b}}, \boldsymbol{\theta}_n^{\mathrm{h}}, \{\mathbf{z}_k\})$ in clients, which requires a more efficient algorithm to aggregate the information of clients and obtain a more robust estimate of $(\boldsymbol{\theta}^{\mathrm{b}}, \boldsymbol{\theta}_n^{\mathrm{h}}, \{\mathbf{z}_k\})$ for the global dataset $\mathcal{S}$. To this end, we propose pFedVMP to solve the optimization problems in (P1) and (P2).

### 4.3 pFEDVMP

We introduce pFedVMP by first presenting the local inference on clients, followed by the global aggregation at the PS.

### 4.3.1 LOCAL INFERENCE

To solve the local problem in eq. (7), client $n$ estimates the local factor $q_n(\boldsymbol{\theta}^{\mathrm{b}}, \boldsymbol{\theta}_n^{\mathrm{h}}, \{\mathbf{z}_k\})$, or the factors $q_n(\boldsymbol{\theta}^{\mathrm{b}})$, $q_n(\boldsymbol{\theta}_n^{\mathrm{h}})$, $q_n(\{\mathbf{z}_k\})$. We alternatively update the factors of model parameters $q_n(\boldsymbol{\theta}^{\mathrm{b}})$, $q_n(\boldsymbol{\theta}_n^{\mathrm{h}})$ and the factor of feature centroids $q_n(\{\mathbf{z}_k\})$ . Specifically, we update the factors of model parameters $q_n(\boldsymbol{\theta}^{\mathrm{b}})$, $q_n(\boldsymbol{\theta}_n^{\mathrm{h}})$ by fixing $q_n(\{\mathbf{z}_k\})$ first. Based on the updated factor $q_n(\boldsymbol{\theta}^{\mathrm{b}})$, we obtain the set of local feature samples $\mathcal{Z}_n$, and update the factor of feature centroid $q_n(\{\mathbf{z}_k\})$.

**Updates the factors** $q_n(\boldsymbol{\theta}^{\mathrm{b}})$ **and** $q_n(\boldsymbol{\theta}_n^{\mathrm{h}})$ Given the problem in (P2), since client $n$ only has a local dataset $\mathcal{S}_n$, it is difficult to sample the joint distribution $p$ directly. Thus, on client $n$, by fixing the cavity factors, we define a surrogate distribution $\tilde{q}_n$ to approximate the joint distribution $p$. The local optimization problem in eq. (7) is converted to

$$(\text{P3}) \quad \min_{\tilde{q}_n(\boldsymbol{\theta}^{\mathrm{b}}\boldsymbol{\theta}_n^{\mathrm{h}},\{\mathbf{z}_k\})} \quad D_{\mathrm{KL}}\left(\tilde{q}_n(\boldsymbol{\theta}^{\mathrm{b}},\boldsymbol{\theta}_n^{\mathrm{h}},\{\mathbf{z}_k\})\|q_n(\boldsymbol{\theta}^{\mathrm{b}},\boldsymbol{\theta}_n^{\mathrm{h}},\{\mathbf{z}_k\})q_{-n}(\boldsymbol{\theta}^{\mathrm{b}})q_{-n}(\{\mathbf{z}_k\})q_{-n}(\{\boldsymbol{\theta}_n^{\mathrm{h}}\}))\right) \tag{8a}$$

$$\text{s.t.} \quad \tilde{q}_n(\boldsymbol{\theta}^{\mathrm{b}},\boldsymbol{\theta}_n^{\mathrm{h}},\{\mathbf{z}_k\})=p(\mathcal{S}_n|\boldsymbol{\theta}^{\mathrm{b}},\boldsymbol{\theta}_n^{\mathrm{h}})q_n(\{\mathbf{z}_k\})q_{-n}(\boldsymbol{\theta}^{\mathrm{b}})q_{-n}(\{\mathbf{z}_k\})q_{-n}(\{\boldsymbol{\theta}_n^{\mathrm{h}}\}), \tag{8b}$$

We now introduce the updates of the factors $q_n(\boldsymbol{\theta}^{\mathrm{b}})$ and $q_n(\boldsymbol{\theta}_n^{\mathrm{h}})$. To solve the problem in (P3), stochastic gradient Markov Chain Monte Carlo (SG-MCMC) is a widely used algorithm to draw samples of $\boldsymbol{\theta}^{\mathrm{b}}, \boldsymbol{\theta}_n^{\mathrm{h}}$ from the distribution $\tilde{q}_n(\boldsymbol{\theta}^{\mathrm{b}},\boldsymbol{\theta}_n^{\mathrm{h}},\{\mathbf{z}_k\})$ (Al-Shedivat et al., 2021; Guo et al., 2023). However, it requires a sufficient number of samples to achieve the factors $q_n(\boldsymbol{\theta}^{\mathrm{b}})$ and $q_n(\boldsymbol{\theta}_n^{\mathrm{h}})$ that approximates the distributions $\tilde{q}_n$ well. This costs an unbearable computational complexity on the client side, and leads to extra communication overhead to upload the covariance matrices of the model parameters $\boldsymbol{\theta}^{\mathrm{b}}$. Thus, we use the traditional SGD method to update the factors $q_n(\boldsymbol{\theta}^{\mathrm{b}})$ and $q_n(\boldsymbol{\theta}_n^{\mathrm{h}})$. The traditional SGD method can be seen as a low-cost implementation of SG-MCMC since the results of $\boldsymbol{\theta}^{\mathrm{b}}, \boldsymbol{\theta}_n^{\mathrm{h}}$ updated by SGD can be regarded as a single sample drawn by SG-MCMC, which reduces the computational and storage cost in the sampling.

Specifically, by taking logarithm on eq. (8b) and drop the terms unrelated to $(\boldsymbol{\theta}_n^{\mathrm{b}}, \boldsymbol{\theta}_n^{\mathrm{h}})$, we minimize the following loss function via SGD:

$$\sum_{i=1}^{S_n}\left(-\log p(\mathbf{x}_{n,i}, y_{n,i}|\boldsymbol{\theta}^{\mathrm{b}}, \boldsymbol{\theta}_n^{\mathrm{h}})+\xi_1\|\mathbf{z}_{n,i}-\boldsymbol{\mu}_{y_{n,i}}^{\mathrm{z}}\|^2\right), \tag{9}$$

where $\boldsymbol{\mu}_{y_{n,i}}^{\mathrm{z}}$ denotes the mean of features in class $y_{n,i}$, $\mathbf{z}_{n,i}$ is the feature corresponding to the data sample $\mathbf{x}_{n,i}$, and $\xi_1$ is a penalty scaler. (The detailed derivation from eq. (8b) to eq. (9) is provided in Appendix A.) Assuming that SGD is performed for the $B_n$ steps on client $n$, we update the mean and the covariance matrix of $q_n(\boldsymbol{\theta}^{\mathrm{b}})$ and $q_n(\boldsymbol{\theta}_n^{\mathrm{h}})$ by

$$\boldsymbol{\mu}_n^{\mathrm{b}} = \boldsymbol{\theta}_n^{\mathrm{b}(B_n)}, \boldsymbol{\Lambda}_n^{\mathrm{b}} = \frac{S_n}{S}\mathbf{I}; \text{ and } \boldsymbol{\mu}_n^{\mathrm{h}} = \boldsymbol{\theta}_n^{\mathrm{h}(B_n)}, \boldsymbol{\Lambda}_n^{\mathrm{h}} = \frac{S_n}{S}\mathbf{I}; \tag{10}$$

where $\boldsymbol{\theta}_n^{\mathrm{b}(B_n)}$ and $\boldsymbol{\theta}_n^{\mathrm{h}(B_n)}$ denote the base model parameter and the head model parameter obtained by client $n$ after $B_n$ steps. We set the covariance matrix as a scaled diagonal matrix proportioned to the size of local datasets for a low implementation cost.

**Updates the factor** $q_n(\{\mathbf{z}_k\})$ We now discuss the factor of the feature centroids $\{\mathbf{z}_k\}$. Based on the GM model defined in eq. (5), the distribution $q_n(\mathbf{z}_k)$ for the $k$-th class is a Gaussian distribution. Thus, for the feature centroid of class $k$, i.e., $\mathbf{z}_{n,i} \in \mathcal{Z}_{k,n}$, the messages of the distribution $q_n(\mathbf{z}_k)$ are estimated by maximize the likelihood of $\{\mathbf{z}_k\}$ with given the based model parameter $\boldsymbol{\theta}^{\mathrm{b}}$ (i.e., the mean $\boldsymbol{\mu}_n^{\mathrm{b}}$) and the local data set $\mathcal{S}_n$, given by

$$\max_{\{(\boldsymbol{\mu}_{k,n}^{\mathrm{z}},\boldsymbol{\Lambda}_{k,n}^{\mathrm{z}})\}} p(\{\mathbf{z}_k\}|\mathcal{S}_n,\boldsymbol{\theta}^{\mathrm{b}}) \Rightarrow \boldsymbol{\mu}_{k,n}^{\mathrm{z}} = \frac{1}{Z_{k,n}}\sum_{i=1}^{Z_{k,n}}\mathbf{z}_{n,i}, \ \boldsymbol{\Lambda}_{k,n}^{\mathrm{z}} = (\boldsymbol{\Sigma}_{k,n}^{\mathrm{z}})^{\dagger}+\alpha\mathbf{I}, \forall k \in [K], \tag{11}$$

where $\boldsymbol{\Sigma}_{k,n}^{\mathrm{z}} = \frac{1}{Z_{k,n}}\sum_{i=1}^{Z_{k,n}}(\mathbf{z}_{n,i}^{\mathrm{z}}-\boldsymbol{\mu}_{k,n})(\mathbf{z}_{n,i}^{\mathrm{z}}-\boldsymbol{\mu}_{k,n})^{\mathsf{T}}$, $(\cdot)^{\dagger}$ denotes the Moore-Penrose inverse, and $\alpha > 0$ is a hyper-parameter to ensure that the precision matrix $\boldsymbol{\Lambda}_{k,n}^{\mathrm{z}}$ is full rank.

### 4.3.2 GLOBAL AGGREGATION

As discussed in Section 3, precise global feature centroids helps to prevent the models from overfitting to local data. Consequently, global aggregation at the PS involves aggregating both the base

model parameters, $\boldsymbol{\theta}^{\mathrm{b}}$, and the local feature centroids, $\mathbf{z}_k$, from the clients. In this subsection, we introduce the distribution aggregation at the PS.

We first introduce the message aggregation of the base model parameters. Let $q(\boldsymbol{\theta}^{\mathrm{b}}) = q_{\mathrm{pri}}(\boldsymbol{\theta}^{\mathrm{b}}) \prod_{n=1}^{N} q_n(\boldsymbol{\theta}^{\mathrm{b}})$ denote the aggregated distribution of $q(\boldsymbol{\theta}^{\mathrm{b}})$. Due to the Gaussian factors $q_{\mathrm{pri}}(\boldsymbol{\theta}^{\mathrm{b}})$, and $q_n(\boldsymbol{\theta}^{\mathrm{b}})$, the aggregated distribution $q(\boldsymbol{\theta}^{\mathrm{b}})$ is also a Gaussian distribution. Based on the product principle of Gaussian distributions, the aggregated messages of $q(\boldsymbol{\theta}^{\mathrm{b}})$ are given by

$$\boldsymbol{\Lambda}^{\mathrm{b}} = \sum_{n=1}^{N} \boldsymbol{\Lambda}_n^{\mathrm{b}}, \text{ and } \boldsymbol{\mu}^{\mathrm{b}} = (\boldsymbol{\Lambda}^{\mathrm{b}})^{-1}\Big(\sum_{n=1}^{N} \boldsymbol{\Lambda}_n^{\mathrm{b}} \boldsymbol{\mu}_n^{\mathrm{b}}\Big). \tag{12}$$

We now discuss the message aggregation of the feature centroids $\mathbf{z}_k$. In the context of supervised learning, the feature centroid $\mathbf{z}_k$ corresponds to class $k$. We assume that the class information of the feature centroids $\{\mathbf{z}_k\}$ is known at the PS beforehand. Thus, the aggregation of $q_n(\mathbf{z}_k)$ is performed on each class $k$ separately, resulting in an aggregation of multiple Gaussian distributions for each class $k$. Specifically, the global distribution of feature centroids $q(\mathbf{z}_k)$ is given by $q(\{\mathbf{z}_k\}) = \sum_{k=1}^{K} q(y_k) q(\mathbf{z}_k)$, where $q(y_k)$ is the component coefficient for class $k$, and $q(\mathbf{z}_k)$ is the distribution of the feature centroid $\mathbf{z}_k$ in class $k$. Based on the product principle of Gaussian distributions, for each class $k$, the mean and the precision matrix of $q(\mathbf{z}_k)$ are given by

$$\boldsymbol{\Lambda}_k^{\mathrm{z}} = \sum_{n=1}^{N} \boldsymbol{\Lambda}_{k,n}^{\mathrm{z}}, \text{ and } \boldsymbol{\mu}_k^{\mathrm{z}} = (\boldsymbol{\Lambda}_k^{\mathrm{z}})^{-1}\Big(\sum_{n=1}^{N} \boldsymbol{\Lambda}_{k,n}^{\mathrm{z}} \boldsymbol{\mu}_{k,n}^{\mathrm{z}}\Big). \tag{13}$$

As for the component coefficient $q(y_k)$, based on the assumption that the PS knows the statistical properties of local datasets, the component coefficient $q(y_k)$ is estimated by $q(y_k) = \frac{\sum_{n=1}^{N} Z_{k,n}}{S}$.

We note that since each factor of distribution $q(\boldsymbol{\theta}^{\mathrm{b}}, \{\boldsymbol{\theta}_n^{\mathrm{h}}\}, \{\mathbf{z}_k\})$ is either Gaussian distribution or GM distribution, the MAP estimate is taking the mean of each Gaussian distribution (or each Gaussian component of the GM distribution), i.e., $\boldsymbol{\mu}_{\mathrm{g}}^{\mathrm{b}}, \{\boldsymbol{\mu}_n^{\mathrm{h}}\}, \{\boldsymbol{\mu}_k^{\mathrm{z}}\}$. We summarize the proposed pFed-VMP in Algorithm 1.

---

**Algorithm 1** `pFedVMP`

**Input:** Local datasets $\{\mathcal{S}_n\}$
1: **for** round $t = 1, \ldots, T$ **do**
2:     **Broadcast** $q(\boldsymbol{\theta}^{\mathrm{b}}, \{\boldsymbol{\theta}_n^{\mathrm{h}}\}, \{\mathbf{z}_k\})$ to clients.
3:     **for** each client $n \in [N]$ **in parallel do**
4:         $q_n(\boldsymbol{\theta}^{\mathrm{b}}), q_n(\boldsymbol{\theta}_n^{\mathrm{h}}), \{q_n(\mathbf{z}_k)\}$
            $\leftarrow$ `LocalInfer`$(q(\boldsymbol{\theta}^{\mathrm{b}}, \{\boldsymbol{\theta}_n^{\mathrm{h}}\}, \{\mathbf{z}_k\}))$
5:     **end for**
6:     **Collect** $q_n(\boldsymbol{\theta}^{\mathrm{b}})$ and $\{q_n(\mathbf{z}_k)\}$ from clients.
7:     $q(\boldsymbol{\theta}^{\mathrm{b}}, \{\boldsymbol{\theta}_n^{\mathrm{h}}\}, \{\mathbf{z}_k\})$
        $\leftarrow$ `GlobalAgg`$(\{q_n(\boldsymbol{\theta}^{\mathrm{b}})\}, \{q_n(\mathbf{z}_k)\})$
8: **end for**
**Output:** $\boldsymbol{\mu}_{\mathrm{g}}^{\mathrm{b}}, \{\boldsymbol{\mu}_n^{\mathrm{h}}\}, \{\boldsymbol{\mu}_k^{\mathrm{z}}\}$

---

**Algorithm 2** `LocalInfer`

**Input:** $q(\boldsymbol{\theta}^{\mathrm{b}}, \{\boldsymbol{\theta}_n^{\mathrm{h}}\}, \{\mathbf{z}_k\})$
1: Update $(\boldsymbol{\mu}_n^{\mathrm{b}}; \boldsymbol{\mu}_n^{\mathrm{h}})$ with performing SGD on the loss function in eq. (9);
2: Update $\{(\boldsymbol{\mu}_{k,n}^{\mathrm{z}}, \boldsymbol{\Sigma}_{k,n}^{\mathrm{z}})\}$ via eq. (11)
**Output:** $q_n(\boldsymbol{\theta}^{\mathrm{b}}), q_n(\boldsymbol{\theta}_n^{\mathrm{h}}), \{q_n(\mathbf{z}_k)\}$

---

**Algorithm 3** `GlobalAgg`

**Input:** $\{q_n(\boldsymbol{\theta}^{\mathrm{b}})\}, \{q_n(\mathbf{z}_k)\}$
1: Compute $(\boldsymbol{\mu}^{\mathrm{b}}, \boldsymbol{\Lambda}^{\mathrm{b}})$ via eq. (12);
2: Compute $(\boldsymbol{\mu}_k^{\mathrm{z}}, \boldsymbol{\Lambda}_k^{\mathrm{z}})$ via eq. (13) for $\forall k \in [K]$;
**Output:** $q(\boldsymbol{\theta}^{\mathrm{b}}; \boldsymbol{\eta}_{\mathrm{g}}^{\mathrm{b}}, \boldsymbol{\Lambda}_{\mathrm{g}}^{\mathrm{b}})$, and $q(\{\mathbf{z}_k\}; \{\boldsymbol{\mu}_k, \boldsymbol{\Sigma}_k\})$

---

## 5 NUMERICAL EXPERIMENT

### 5.1 SETUP

**Baselines, datasets and backbones.** We compare the performance of pFedVMP with the following state-of-the-art pFL algorithms: FedAvg-FT where the global model is fine-tuned locally on each client; FedRep, FedPer, FedROD; FedProto, MOON, FedCP, GPFL, FedPAC; FedPA-FT, FedEP-FT, QLSD-FT, pFedGP, pFedBreD. The hyperparameters of the baselines are set according to the original papers. We use a 4-layer convolution neural network for FMNIST (Xiao et al., 2017), EMNIST (Cohen et al., 2017), and Cifar10/Cifar100 (Krizhevsky et al., 2009). The details of the CNN architecture are presented in Appendix B.

**Data heterogeneous settings.** Based on the above datasets, following Lin et al. (2020), we consider the following data heterogeneous setting: Let $q_{k,n} = \frac{Z_{k,n}}{S_k}$ denote the proportion of data samples

from class $k$ allocated to client $n$, and let $\mathbf{q}_k = [q_{k,1}, \ldots, q_{k,N}]$ denote the proportion values for class $k$ across all clients. Naturally, $\sum_{n=1}^{N} q_{k,n} = 1$. For each class $k$, the entries of $\mathbf{q}_k$ are sampled from a Dirichlet distribution, denoted by $\mathrm{Dir}(\beta)$, where $\beta$ is the parameter of the Dirichlet distribution. A small $\beta$ leads to a greater concentration of data from the same class in a few clients.

**Implementation details.** We consider a scenario that all clients participate in FL training. On each client, the local dataset is divided into 80% for training and 20% for testing. We set $\alpha = 1$. A total of 1000 communication rounds are conducted between the PS and clients, with one local epoch per round. The SGD optimizer is used to update both the base model and the head models, with a learning rate of 0.01 and a batch size of 10. We report the mean values across three trials.

## 5.2 RESULTS

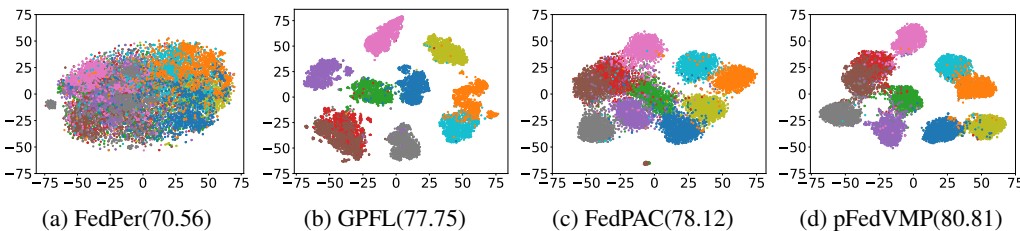

| (a) FedPer(70.56) | (b) GPFL(77.75) | (c) FedPAC(78.12) | (d) pFedVMP(80.81) |

Figure 2: The t-SNE visualization results of feature vectors obtained by pFedVMP and other FL algorithms. We consider $50$ clients on Cifar10. The test accuracy is reported behind each subtitle.

**Learned Features.** We first visualize the feature samples. We train 50 clients on the CIFAR-10 dataset, partitioning each class of data among the clients according to $\mathrm{Dir}(0.3)$. In Fig. 2, we plot the low-dimensional representation of the high-dimensional features using t-SNE (Van der Maaten & Hinton, 2008), where each color represents a class, and each point corresponds to a feature sample. Due to the limited data available to each client, a base model overfitting to local data will project the data in the same class into distinct clusters. In contrast, a base model with stronger generalization tends to project data within the same class into a single cluster, as modeled by the GM model in eq. (5). Moreover, the more distinct the data from different classes, the easier it becomes to learn robust personal classifier heads. From Fig. 2(a), we see that the base model learned by FedPer projects the data into the same cluster, resulting in a poor classification performance. By adding the constraints to the output features, GPFL, FedPAC, and pFedVMP achieves better values of test accuracy. Although GPFL discriminates the features of data from different classes, the features from different clients exhibit greater divergence and form some stragglers from the centroid, indicating that the base model in GPFL overfits the local data. Compared to pFedVMP, the boundary of the features from different classes obtained by FedPAC are not discriminative to each other, resulting in worse performance. This is because the feature centroid aggregation method used in FedPAC is based on weighted average, leading to a larger covariance of the features within each class. As shown in Fig. 2d, features within the same class are closely grouped and tend to form a hyper-oval shape, distancing themselves from other classes, which validates the GM model in eq. (5). This result demonstrates that pFedVMP achieves a better balance between generalization and personalization.

**Effectiveness.** We now compare pFedVMP with other SOTA baselines. We report the test accuracy values averaged on the clients obtained by the algorithms in Table 1. We also plot the average test accuracy and training loss of various pFL algorithms in Fig 1. The average test accuracy is given by $\frac{\sum_{n=1}^{N} A_n^{\mathrm{c}}}{\sum_{n=1}^{N} A_n}$, where $A_n$ denotes the number of test data on client $n$, and $A_n^{\mathrm{c}}$ denotes the number of correct classified data on client $n$. Here, we consider two data partition settings, $\mathrm{Dir}(0.1)$ and $\mathrm{Dir}(0.3)$, where data samples are more concentrated on a few clients in $\mathrm{Dir}(0.1)$, and the local data for each client come from more classes in $\mathrm{Dir}(0.3)$. As shown in Table 1 and Fig 3, pFedVMP achieves the highest test accuracy in the various settings, demonstrating the superior performance of pFedVMP. Next, we explain the reasons for the superior performance of pFedVMP over other baseline methods based on the experimental results. (1) **pFed-VMP v.s. FedAvg-FT**: FedAvg-FT forces the model on each client aligned to the global model at the PS, which prevents the model from overfitting the local data and results in competitive performance. However, FedAvg-FT does not involve the constraints on the features, performing

worse than pFedVMP. (2) **pFedVMP v.s. FedPer & FedRep & FedROD**: The baselien methods, FedPer, FedRep and FedROD, train a base model to extract the features without regularizing the learned features to concentrate to global feature centroids. By adding this constraint, pFedVMP outperforms FedPer/FedRep/FedROD by 11.38%/8.97%/8.13% on Cifar100 in $\text{Dir}(0.1)$.

(3) **pFedVMP v.s. FedProto & MOON & FedCP & GPFL & FedPAC**: These algorithms guide feature extraction with global feature centroids. FedProto does not share the local base model, causing the base model to suffer from overfitting on the local data. As shown in Fig 2, although GPFL shares the base model, the features of the same class still diverge from the global feature centroid, resulting in a poor performance. In FedPAC, the boundary of feature samples from different classes are not distriminative from each other due to the weighted average aggregation of the feature centroids. By sharing the base model and aggregating the distributions of global feature centroids, pFedVMP outperforms FedProto/MOON/FedCP/GPFL/FedPAC by 14.10%/3.05%/7.85%/3.06%/2.69% on Cifar10 in $\text{Dir}(0.3)$.

(4) **pFedVMP v.s. FedPA-FT & FedEP-FT & QLSD-FT & pFedGP & pFedBreD**: These BFL methods update the model parameters with Bayesian methods. FedPA, FedEP, and QLSD formulate model training as Bayesian inference tasks and aggregate the distributions of local parameters. pFedGP trains personalized Gaussian process classifiers, while pFedBreD injects the personalized prior of model parameters in training. However, they do not leverage the global feature centroids to guide local model training. In constrast, pFedVMP achieves more precise estimates of the distributions of global feature centroids and model parameters by variational message passing.

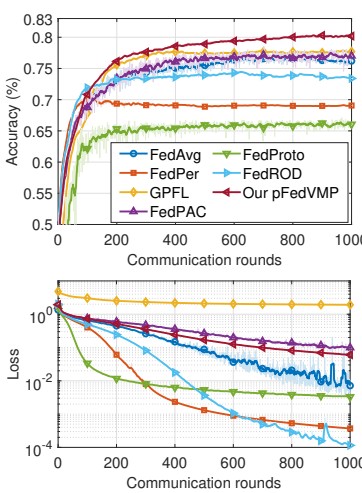

Figure 3: **Upper**: Test accuracy of different pFL algorithms versus communication rounds on Cifar10-50c with $\text{Dir}(0.1)$. **Lower**: Training loss of different pFL algorithms versus communication rounds under the same setting.

Table 1: Comparison of testing accuracy. The highest accuracy results ($\%, \uparrow$) are highlighted in **bold**, while the second highest results are underlined. The values (mean) represent the mean of values from three independent runs. "20c" means the number of clients $N = 20$.

| | FMNIST-50c | | EMNIST-50c | | Cifar10-50c | | Cifar100-20c | |
|---|---|---|---|---|---|---|---|---|
| Distribution | Dir(0.1) | Dir(0.3) | Dir(0.1) | Dir(0.3) | Dir(0.1) | Dir(0.3) | Dir(0.1) | Dir(0.3) |
| FedAvg-FT | 96.99 | 94.93 | 95.95 | 93.46 | 87.93 | 77.70 | 59.41 | 50.79 |
| FedPer | 96.43 | 92.99 | 94.66 | 90.51 | 85.05 | 70.56 | 52.94 | 39.74 |
| FedRep | 96.62 | 93.30 | 94.60 | 90.48 | 85.98 | 70.85 | 55.35 | 41.38 |
| FedROD | 96.68 | 94.38 | 95.54 | 92.66 | 86.35 | 74.61 | 56.19 | 46.42 |
| FedProto | 96.06 | 92.19 | 93.38 | 90.30 | 83.05 | 66.71 | 43.77 | 36.68 |
| MOON | 96.57 | 94.80 | 95.93 | 93.31 | 87.88 | 77.76 | 58.82 | 50.19 |
| FedCP | 96.87 | 93.87 | 95.95 | 92.81 | 86.97 | 72.96 | 59.90 | 47.96 |
| GPFL | 96.65 | 95.09 | 96.71 | 94.82 | 84.80 | 77.75 | 62.50 | 52.48 |
| FedPAC | 96.59 | 94.57 | 96.78 | 94.79 | 87.34 | 78.12 | 63.12 | 55.88 |
| FedPA-FT | 96.91 | 94.97 | 96.36 | 94.20 | 87.88 | 78.23 | 60.32 | 51.67 |
| FedEP-FT | 96.88 | 94.95 | 96.31 | 94.23 | 87.87 | 78.36 | 60.31 | 51.92 |
| QLSD-FT | 93.80 | 89.30 | 91.56 | 87.80 | 79.49 | 65.35 | 37.44 | 27.74 |
| pFedGP | 96.11 | 94.15 | 94.77 | 91.02 | 85.88 | 75.88 | 57.32 | 46.53 |
| pFedBreD | 96.64 | 94.21 | 95.66 | 93.06 | 86.39 | 74.42 | 54.37 | 44.89 |
| pFedVMP | **97.23** | **95.60** | **96.97** | **95.09** | **88.12** | **80.81** | **64.32** | **56.75** |

**Ablation Study.** We conduct an ablation study to further evaluate the efficacy of the feature centroid aggregation proposed in pFedVMP. We compare pFedVMP with the following baselines: (1) pFedVMP-avg, where the local feature centroids $\{\boldsymbol{\mu}_{k,n}^{z}\}$ are aggregated at the PS using a weighted average based on local dataset sizes; and (2) FedPer, where no regularization is applied to the feature

Table 2: The test accuracy (%) of pFedVMP and its degrade versions on Cifar10-50c

|          | pFedVMP | pFedVMP-avg | FedPer |
|----------|---------|-------------|--------|
| Dir(0.1) | 88.12   | 87.29       | 85.05  |
| Dir(0.3) | 80.81   | 76.97       | 70.56  |

centroids. As shown in Table 2, both pFedVMP-avg and pFedVMP outperform FedPer significantly due to the incorporation of constraints on the feature centroids. Moreover, pFedVMP improves the average test accuracy over pFedVMP-avg by producing more discriminative feature representations from different classes, demonstrating the effectiveness of aggregating the distributions of global feature centroids in pFedVMP.

Table 3: The fairness, measured by the coefficient of variation $(\times 10^{-2}, \downarrow)$, of test accuracy across clients' local datasets when achieving the best test accuracy on FMNIST, EMNIST, Cifar10 and Cifar100 in $\text{Dir}(0.3)$. The standard deviation $(\%, \downarrow)$ is presented in blankets.

| Method    | FMNIST-50c  | EMNIST-50c | Cifar10-50c  | Cifar100-20c |
|-----------|-------------|------------|--------------|--------------|
| FedAvg-FT | 4.46(4.23)  | 2.80(2.62) | 12.73( 9.89) | 6.95(3.53)   |
| FedPer    | 6.73(6.26)  | 3.16(2.86) | 19.05(13.44) | 6.47(2.57)   |
| FedROD    | 4.79(4.52)  | 2.97(3.21) | 15.99(11.93) | 7.56(3.51)   |
| FedProto  | 6.91(6.37)  | 3.26(2.94) | 23.58(15.73) | 10.05(3.67)  |
| GPFL      | 4.26(4.06)  | 2.76(2.62) | 13.84(10.76) | 6.00(3.16)   |
| FedPAC    | 5.15(4.87)  | 2.83(2.68) | 14.07(10.99) | 5.87(3.30)   |
| pFedVMP   | **4.14**(3.96) | **2.73**(2.60) | **11.81**( 9.45) | **5.84**(3.33) |

**Fairness Analysis.** We now analyze the fairness of the models obtained by pFedVMP. As discussed in Zhang et al. (2023a), Li et al. (2021b), some clients may perform poorly in the pFL although the average test accuracy is improving. Thus, the fairness of a pFL method is also an important metric. Following Li et al. (2021b), we use the coefficient of variation to measure the fairness of the pFL models, where a smaller coefficient of variation represents a more fair pFL model across the clients. As shown in Table 3, our pFedVMP outperforms other pFL baselines by achieving a much smaller coefficient of variation, especially on Cifar10 with 50 clients, demonstrating the superior performance of pFedVMP.

## 6 CONCLUSIONS

In this paper, we introduced pFedVMP, a novel pFL approach designed to address the challenge of statistical heterogeneity in FL. By leveraging variational message passing, pFedVMP effectively aggregates the distributions of model parameters and feature centroids, enabling precise estimates of their probabilistic models. Our method strikes a balance between incorporating global information for collaborative learning and maintaining personalized models tailored to each client's local dataset. Moreover, pFedVMP effectively mitigates the risk of overfitting through the utilization of global feature centroids to regularize local training. Numerical results demonstrate that pFedVMP outperforms state-of-the-art algorithms in terms of test accuracy and the coefficient of variation.

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

## A DERIVATION THE LOSS FUNCTION IN EQUATION 9

We now derive the loss function of SGD in eq. (9). Based on the above definition of $\tilde{q}_n(\boldsymbol{\theta}^{\mathrm{b}}, \boldsymbol{\theta}_n^{\mathrm{h}}, \{\mathbf{z}_k\})$, the negative logarithm of the target distribution is expressed as:

$$-\log \tilde{q}_n(\boldsymbol{\theta}^{\mathrm{b}}, \boldsymbol{\theta}_n^{\mathrm{h}}, \{\mathbf{z}_k\}) = -\log p(\mathcal{S}_n | \boldsymbol{\theta}^{\mathrm{b}}, \boldsymbol{\theta}_n^{\mathrm{h}}) - \log q(\{\mathbf{z}_k\}) - \log q_{-n}(\boldsymbol{\theta}^{\mathrm{b}}) - \log q_{-n}(\{\boldsymbol{\theta}_n^{\mathrm{h}}\}) + \mathrm{Const}.$$

On client $n$, computing the cavity factors $q_{-n}(\boldsymbol{\theta}^{\mathrm{b}})$ and $q_{-n}(\{\boldsymbol{\theta}_n^{\mathrm{h}}\})$ may lead to instability during sampling. Thus, we exclude the terms involving $q_{-n}(\boldsymbol{\theta}^{\mathrm{b}}), q_{-n}(\{\boldsymbol{\theta}_n^{\mathrm{h}}\})$, resulting in the following simplified loss function:

$$-\log p(\mathcal{S}_n | \boldsymbol{\theta}^{\mathrm{b}}, \boldsymbol{\theta}_n^{\mathrm{h}}) - \log q(\{\mathbf{z}_k\})$$

By assuming the data samples are i.i.d., we obtain eq. (9):

$$\sum\nolimits_{i=1}^{S_n} \left( -\log p(\mathbf{x}_{n,i}, y_{n,i} | \boldsymbol{\theta}^{\mathrm{b}}, \boldsymbol{\theta}_n^{\mathrm{h}}) + \xi_1 \| \mathbf{z}_{n,i} - \boldsymbol{\mu}_{y_{n,i}}^{\mathrm{z}} \|^2 \right), \tag{14}$$

where the second term is because calculating the precision matrix $\boldsymbol{\Lambda}_{y_{n,i}}^{\mathrm{z}}$ in the loss function may cause the gradient unstable, and we use a spherical Gaussian distribution with the mean $\boldsymbol{\mu}_{y_{n,i}}^{\mathrm{z}}$ and the precision matrix $\xi_1 \mathbf{I}$ instead.

## B DETAILS OF EXPERIMENTAL SETUP

**Hardware Information.** We implement all the FL baselines and the proposed pFedVMP algorithm with PyTorch and simulate them with NVIDIA GeForce RTX 2080Ti GPUs.

**Dataset.** We use the FMNIST (Xiao et al., 2017), EMNIST-balanced (Cohen et al., 2017), and CIFAR-10/CIFAR-100 (Krizhevsky et al., 2009) datasets in our experiments. For each dataset, we uniformly sample from the entire dataset to construct a new subset. Specifically, the retained proportions are $25\%$ for FMNIST-50c-Dir(0.3) and CIFAR-10-50c-Dir(0.3), $50\%$ for FMNIST-50c-Dir(0.1) and CIFAR-10-20c-Dir(0.1), and $100\%$ for EMNIST-50c-Dir(0.1), EMNIST-50c-Dir(0.3), CIFAR-100-20c-Dir(0.1), and CIFAR-100-20c-Dir(0.3).

**Data Heterogeneity Setting.** Following prior work in pFL (Lin et al., 2020; Zhang et al., 2023a), we generate local datasets for clients based on a Dirichlet distribution. Specifically, let $q_{k,n} = \frac{Z_{k,n}}{S_k}$ represent the proportion of data samples from class $k$ allocated to client $n$, and let $\mathbf{q}_k = [q_{k,1}, \ldots, q_{k,N}]$ denote the proportion values for class $k$ across all clients, where $\sum_{n=1}^{N} q_{k,n} = 1$. For each class $k$, the entries of $\mathbf{q}_k$ are sampled from a Dirichlet distribution, denoted by $\mathrm{Dir}(\beta)$, where $\beta$ is the distribution parameter. A smaller $\beta$ results in a higher concentration of data from the same class within a few clients. The data distribution is visualized in Fig.4. As shown in Fig.4, data for each class are more concentrated among a few clients when $\mathrm{Dir}(0.1)$ is used compared to $\mathrm{Dir}(0.3)$. In contrast, in the case of $\mathrm{Dir}(0.3)$, each client contains a greater variety of data categories than in the case of $\mathrm{Dir}(0.1)$.

**Network Architecture.** We present the architecture of the CNN used in our experiments in Table 4. Some parameters of the network, including `data_channels`, `dim`, and `class_num`, vary across datasets and are listed in Table 4.

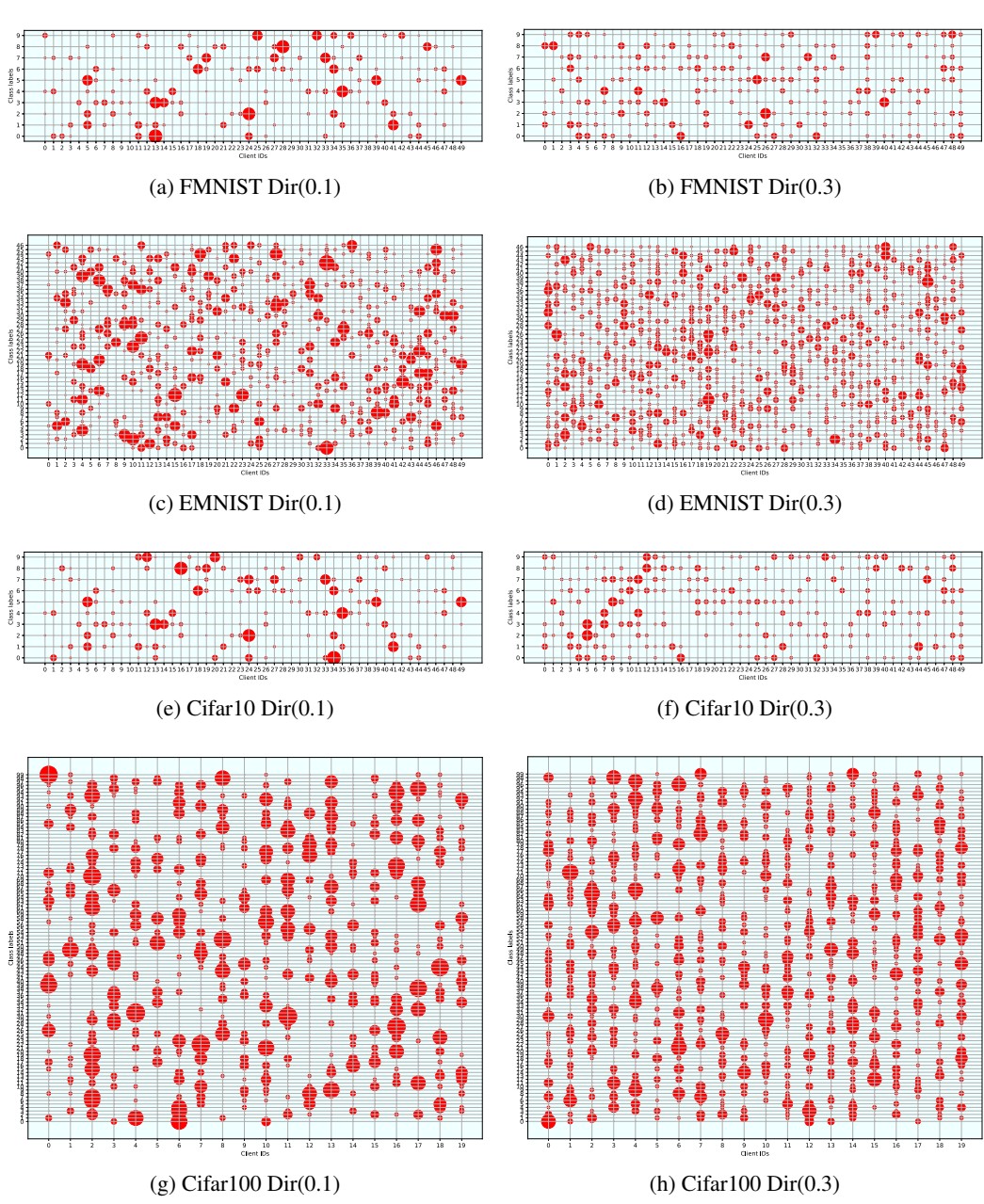

Figure 4: The bubble charts for visualizing the data distributions. Each row represents the distribution of data with the same label across clients, while each column indicates the data partitioned to a specific client. The size of the bubble corresponds to the relative size of the local dataset, with larger bubbles representing more data.

Table 4: The architecture of the CNN used in the experiments.

| Layer type | Layer details |
|---|---|
| Conv2d | in_channels=`data_channels`, out_channels=32, kernel_size=5, stride=1, padding=0 |
| LeakyReLU | negative_slope=0.1, inplace=True |
| MaxPool2d | kernel_size=2x2 |
| Conv2d | in_channels=32, out_channels=64, kernel_size=5, stride=1, padding=0 |
| LeakyReLU | negative_slope=0.1, inplace=True |
| MaxPool2d | kernel_size=2x2 |
| Flatten | - |
| Linear | in_features=`dim`, out_features=512 |
| LeakyReLU | negative_slope=0.1, inplace=True |
| Linear | in_features=512, out_features=`class_num` |

| Dataset | Parameters details |
|---|---|
| FMNIST | `data_channels` = 1, `dim` = 1024, `class_num` = 10 |
| EMNIST | `data_channels` = 1, `dim` = 1024, `class_num` = 47 |
| CIFAR10 | `data_channels` = 3, `dim` = 1600, `class_num` = 10 |
| CIFAR100 | `data_channels` = 3, `dim` = 1600, `class_num` = 100 |

## C ADDITIONAL EXPERIMENTAL RESULTS

### C.1 RESULTS IN PATHOLOGICAL NON-I.I.D. DATA SCENARIO

To evaluate various non-i.i.d. data scenarios, we follow Shamsian et al. (2021); Zhang et al. (2023a); Xu et al. (2023) and present results on the pathological non-i.i.d. data distribution. In this scenario, local datasets are small, and FL models are at high risk of overfitting. Specifically, using Cifar10 as a benchmark, we select 3 different classes for each client and randomly sample 100 instances from each class. The number of clients is set to 50. We compare pFedVMP with the other baseline methods under the pathological non-i.i.d. data distribution and report the test accuracy of the methods in Table 5. As shown in Table 5, pFedVMP still achieves the best average test accuracy than other pFL baselines thanks to its more precise estimation of global feature centroids and model parameters based on message passing, which demonstrates the superb performance of pFedVMP in the scenario of pathological non-i.i.d. data.

Table 5: The test accuracy ($\%, \uparrow$) under pathological Non-i.i.d. data distributions on Cifar10.

| Methods | FedAvg-FT | FedPer | FedROD | FedProto | GPFL | FedPAC | pFedVMP |
|---|---|---|---|---|---|---|---|
| Test accuracy | 83.47 | 74.80 | 79.07 | 71.83 | 82.63 | 81.17 | **84.73** |

### C.2 EFFECT OF FEATURE DIMENSIONS

In representation learning, the dimensionality of the feature space is an important hyperparameter, closely related to model capacity and overfitting risk (Goodfellow et al., 2016; Alain, 2016). Due to the essential role of global feature centroids, we investigate the effect of feature dimensions on pFedVMP here, denoted by `dim` as shown in Appendix B. We report the test accuracy of pFedVMP across different feature dimensions on Cifar10 and Cifar100 datasets in Table 6. As shown in Table 6, the best test accuracy is at 256 for Cifar10 and 640 for Cifar100. The explanations are given as follows. Increasing feature dimensions enhances model capacity, thereby improving the learning performance of FL models. However, it also raises the number of trainable parameters, which increases the risk of overfitting. In the FL context, where some clients have small local datasets, this risk is mitigated.

Table 6: The test accuracy (%, ↑) of pFedVMP under different feature dimensions.

|  | 128 | 256 | 384 | 512 | 640 |
|---|---|---|---|---|---|
| Cifar10-50c-Dir0.3 | 79.46 | 80.14 | 80.10 | 79.93 | 79.82 |
| Cifar100-20c-Dir0.3 | 52.04 | 55.33 | 56.79 | 56.81 | 57.19 |

## C.3 EFFECT OF PENALTY SCALAR $\xi_1$

In this subsection, we investigate the effect of the penalty scalar $\xi_1$ on the learning performance of pFedVMP. We evaluated a range of $\xi_1$ values on the scenario of Cifar10-50c $\mathrm{Dir}(0.3)$ and present the average test accuracy in Table 7. Table 7 shows the varying learning performance of pFedVMP under different values of $\xi_1$ in eq. (9). The best value of $\xi_1$ is $50$ in this scenario. With the increasing of $\xi_1$, the average test accuracy improves first from $\xi_1 = 1$ to $\xi_1 = 50$ but declines as $\xi_1$ rises to 100. This behavior arises because a smaller $\xi_1$ weakens the regularization term on local updates, increasing the possibility of local models overfitting the data. Conversely, a larger $\xi_1$ restricts the model's ability to explore, resulting in a suboptimal performance.

Table 7: The test accuracy (%, ↑) of pFedVMP under different values of penalty scalar $\xi_1$.

| $\xi_1$ | 1 | 5 | 10 | 20 | 50 | 70 | 100 |
|---|---|---|---|---|---|---|---|
| Test accuracy | 75.15 | 78.66 | 79.22 | 80.08 | 80.81 | 80.25 | 79.43 |

