# OpenReview forum: "Personalized Federated Learning via Variational Massage Passing"
_ICLR.cc/2025/Conference — Submitted to ICLR 2025_

### Official Review · Reviewer_g6Ef · 2024-11-03

**Soundness:** 3
**Presentation:** 2
**Contribution:** 2
**Rating:** 6
**Confidence:** 4

**Summary:**

- This paper presents a personalizd federated learning methods, called pFedVMP, as a new solution to tailed personalized model for local clients. The core idea is to model both the centorids of parameters and features.

**Strengths:**

- The combination of modeling both feature space and parameter space seems to be good as shown in the reported results of the experiments.

**Weaknesses:**

- The main ideas of pFedVMP, modeling parameters and feature centroids, are not new. A Bayesian perspective is either an well-explored areas in federated learning.
- Even tough this area is well-explored in recent years, most of the baselines in the experiments are not the newest, which make it hard to believe to be SOTA. Moreover, the most related works are not included as a baseline of Bayesian Federated Learning.
    1. Related and new baselines are recommended as followings: feature modeling [1,2] and parameter modeling [3,4].
    2. MOON [1] has similar claims about feature centroids modeling, and however is not discussed.
    3. PRIOR [4] emphasizes the importance of global prior information which is the parameter centroid, which has not been discussed yet.
    4. More comparison about works after year 2023 should be added beyond FedPAC (ICLR 2023) in order to be claimed as SOTA.
- Some words, e.g., leveraging second-order statistical information, are confusing in the federated learning. The ambiguous words in this comprehensive field can refer to second-order moments, covariance matrices, or second-order gradients, Hessian matrices.

[1] Model-contrastive federated learning. CVPR 2021

[2] FedCP: Separating Feature Information for Personalized Federated Learning via Conditional Policy. KDD 2023

[3] Personalized Federated Learning via Variational Bayesian Inference. ICML 2022

[4] PRIOR: Personalized Prior for Reactivating the Information Overlooked in Federated Learning.  NeurIPS 2023

**Questions:**

- What's the main difference between the propoed pFedVMP and the methods use GMM to model the feature centroids or parameter centeroids?
- The main claim, the global feature centroids is important, is already a common sense in the literature of federated learning, which is first systematically claimed and proven by MOON [1] as far as I know. What's more?
- If the difference is clearly explained, a positive rating is considered.

[1] Model-contrastive federated learning. CVPR 2021

---

> ### Author Response · Authors · 2024-11-28
>
> **We thank Reviewer g6Ef for careful reading and constructive comments.**
>
> **W1 Difference between pFedVMP and other Bayesian federated learning models**
>
> BFL can be broadly categorized into client-side BFL and server-side BFL in terms of FL architectures [R1]. Client-side BFL focuses on learning Bayesian local models on client nodes, while server-side BFL aggregates local updates for global models using Bayesian methods. Our paper belongs to server-side BFL.
>
> The prior server-side BFL works related to our paper include FedPA, FedEP, QLSD, pFedBayes, which formulate model training as Bayesian inference tasks and aggregate the distributions of local **parameters**. FedPA (Al-Shedivat et al., 2021) approximated the posterior distribution into the product
> of distributions with respect to local datasets during local model training. FedEP (Guo et al., 2023)
> developed the Bayesian model aggregation rule by using expectation propagation. QLSD (Vono
> et al., 2022) extended the approach in FedPA with the quantized Langevin stochastic dynamics for
> local update. pFedBayes (Zhang et al., 2022) uses variational inference to approximate the posterior distribution of local model parameters for each client, and aggregates local models on the server.
>
> However, the above BFL methods do not utilize **global feature centroids** to guide local model training, which limits their ability to effectively address data heterogeneity. In contrast, pFedVMP considers both model **parameters** and **feature** centroids. The gain of pFedVMP comes from the following two folds: on one hand, pFedVMP guides the local training using a **regular** term of global feature centroids, decreasing the over-fitting risk of local training; on the other hand, pFedVMP achieves more **precise estimates** of the distributions of global feature centroid by variational message passing.
>
> [R1] Bayesian Federated Learning: A Survey, IJCAI-23
>
> **W2** Thanks for the constructive comment. As suggested by the reviewer, we have conduct the numerical experiments of comparing the proposed pFedVMP with the baselines. However, as the official implementation of pFedBayes is not publicly available, it was not included in the comparison.
>
> |Dataset|FMNIST(Dir(0.1))|FMNIST(Dir(0.3))|EMNIST(Dir(0.1))|EMNIST(Dir(0.3))|Cifar10(Dir(0.1))|Cifar10(Dir(0.3))|Cifar100(Dir(0.1))|Cifar100(Dir(0.3))|
> |-------------|--------------------|--------------------|--------------------|--------------------|--------------------|--------------------|---------------------|---------------------|
> |MOON|96.57|94.8|95.93|93.31|87.88|77.76|58.82|50.19|
> |FedCP|96.87|93.87|95.95|92.81|86.97|72.96|59.90|47.96|
> |PRIOR|96.64|94.21|95.66|93.06|86.39|74.42|54.37|44.89|
> |**pFedVMP**|**97.23**|**95.60**|**96.97**|**95.09**|**88.12**|**80.81**|**64.22**|**56.75**|
>
>
> As shown in the table above, pFedVMP outperforms the baseline methods under various data partition settings. Although PRIOR leverages personalized prior knowledge, it does not utilize global representation information to guide local training. In contrast, both MOON and FedCP incorporate global representation information; however, MOON uses the similarity between local and global representations, whereas FedCP employs a conditional policy. Thanks to VMP, pFedVMP yields more precise estimates of the distributions of model parameters and global feature centroids, leading to its superior performance.
>
> In the revised manuscript, we have cited these works.
>
> **W3** Thanks for the constructive comment. We understand that this phrase can be interpreted in various ways within the federated learning domain. In our work, "second-order statistical information" specifically refers to covariance matrices derived from the local feature centroids and local model parameters. To ensure clarity, we have revised the manuscript to explicitly define this term and eliminate potential ambiguity.

---

> ### Author Response · Authors · 2024-11-28
>
> **Q1 Comparison with other GMM-based models**
>
> In the following, we discuss the differences between the proposed pFedVMP and the GMM-based methods on the aspects of the feature centroids and model parameter.
>
> **Feature centroids** To our best knowledge, the research of modeling the feature centroids with GMM in pFL is rare. Here, we refer to the works on centralized learning, e.g., DMVCVAE [R2].
>
> - The methods using GMM to model the feature centroids are typically in unsupervised learning. In DMVCVAE uses GMM to model feature centroids for clustering tasks, primarily aiming to train a deep autoencoder and learn shared latent representations that group data points based on similarity.
> - In this paper, we consider supervised learning. At the local training, each feature sample obtained by the base model is forced to be close to the global centroid of its class. At the PS, we assume that the class information of the feature centroids is known beforehand. Thus, the aggregation of feature centroids distributions is performed on each class separately, resulting in an aggregation of multiple Gaussian distributions for each class.
>
> **Model parameter** We consider FedGMM is one of "the methods that use GMM to model the parameter centroids". FedGMM uses GMM to model the distributions of the local data and the model parameters. The local model parameters is a weighted sum of a branch of model parameters. pFedVMP uses Gaussian distributions to model the distribution of the model parameters, and the local model parameters is the mean of the Gaussian distribution. Compared with pFedVMP, FedGMM requires more computational and communication cost to update the multiple components of the model parameters.
>
> [R2] Shared Generative Latent Representation Learning for Multi-View Clustering. AAAI 2020
>
> [R3] Personalized federated learning under mixture of distributions. ICML 2023
>
> **Q2 Difference between pFedVMP and MOON**
>
> We agree with the reviewer that the global feature centroids is important in pFL. Below, we outline the key differences between pFedVMP and MOON.
>
> - **Bayesian modeling** MOON aims to align the features obtained by the local model and the global model on a **sample-wise basis**. In contrast, pFedVMP treats feature centroids as random variables and aggregates their distributions using a maximum-a-posteriori (MAP) criterion. By accounting for both the mean and covariance of feature centroids, pFedVMP provides more **precise estimates** of the global feature centroid distributions, thereby enhancing model performance. To the best of our knowledge, this work is the first to estimate global feature centroids via variational message passing in pFL.
> - **Feature aggregation** MOON focuses on achieving agreement between the local and global model representations on a sample-wise basis and does not aggregate feature centroids at the server. Conversely, pFedVMP aggregates the distributions of feature centroids at the server to achieve more **accurate estimates** of global feature centroids.
> - **Feature alignment** MOON aligns the local and global representations by maximizing their similarity, whereas pFedVMP employs a regularization term involving the global centroids to achieve alignment.
>
> Numerical results shows that the proposed pFedVMP outperforms MOON on the test learning accuracy, highlighting the effectiveness of pFedVMP.
>
> **Q3** Thanks for the constructive comments from the reviewer. We have carefully explained the differences between pFedVMP and the prior BFL or pFedVMP and MOON. We hope this revision ensures the distinction is now clear and supports a positive evaluation.

---

> ### Comment · Reviewer_g6Ef · 2024-11-29
>
> An additional suggestion: experiments should be considered to straightforward support your claim are about how precise the centroids pFedVMP estimates. A better accuracy is not enough, because better performance does not mean less biased estimation of the components in the optimization process.
>
> Most of the main concerns are resolved. I raise my rating accordingly.

---

> > ### Author Response · Authors · 2024-11-30
> >
> > **We sincerely thank Reviewer g6Ef for their positive feedback and for raising their rating.**
> >
> > We also appreciate this constructive suggestion and will consider conducting additional experiments in the future.

---

### Official Review · Reviewer_T9rb · 2024-11-04

**Soundness:** 2
**Presentation:** 3
**Contribution:** 2
**Rating:** 5
**Confidence:** 3

**Summary:**

pFedVMP is a personalized federated learning approach that uses variational message passing to enhance feature aggregation, yielding more precise model parameter estimates and improving training accuracy and fairness under heterogeneous data conditions.

**Strengths:**

This paper provides a nice numerical study with SOTA baselines with interpretation, ablation study, and fairness analysis.

**Weaknesses:**

First of all, there is a typo in the title of this paper: "Massage Passing" should be "Message Passing"...

W1. The Bayesian benchmark models are not included in the comparison.

W2. The computational cost seems to be high, especially with high-dimensional features.

W3. The selection of hyperparameters needs justification.

W4. There is a lack of theoretical guarantees for the proposed method.

**Questions:**

Q1. The paper presents extensive comparisons with various methods in federated representation learning but fails to include benchmark models within the framework of Bayesian federated learning, for example, BNFed, pFedGP, pFedBayes, FedPA, FedEP, QLSD, and others. This weakens the argument for the superiority of the proposed method in the Bayesian context.

Q2. The pFedVMP algorithm involves numerous matrix inversions in each communication round (especially in Equation 12), which can lead to a significant computational burden, particularly with high-dimensional features. It is essential to evaluate the computational cost relative to other methods and propose reasonable solutions to mitigate these costs.

Q3. The algorithm contains several hyperparameters, such as those in Equations 8 and 10. A more in-depth study on the impact of these hyperparameters and a clear justification for their selection is necessary.

Q4. There is a lack of theoretical guarantees for the proposed method.

---

> ### Author Response · Authors · 2024-11-28
>
> **We thank Reviewer T9rb for careful reading and constructive comments.**
>
> **Typographical Error**: We sincerely apologize for the typographical error in the title and will correct "Massage Passing" to "Message Passing" in the revised manuscript.
>
> **W1/Q1 Bayesian benchmarks**: Thanks for this constructive comment. As suggested by the reviewer, we conducted experiments to compare the proposed pFedVMP with the benchmark methods for Bayesian federated learning mentioned by the reviewer, including FedPA, FedEP, QLSD, and pFedGP. Since the official implementation of pFedBayes is not publicly available, and BNFed is applicable only to simple feedforward neural networks, these methods were not included in the comparison. The results are presented in the following table.
>
> |Dataset|FMNIST(Dir(0.1))|FMNIST(Dir(0.3))|EMNIST(Dir(0.1))|EMNIST(Dir(0.3))|Cifar10(Dir(0.1))|Cifar10(Dir(0.3))|Cifar100(Dir(0.1))|Cifar100(Dir(0.3))|
> |-------------|--------------------|--------------------|--------------------|--------------------|--------------------|--------------------|---------------------|---------------------|
> |FedPA-FT|96.91|94.97|96.36|94.20|87.88|78.23|60.32|51.67|
> |FedEP-FT|96.88|94.95|96.31|94.23|87.87|78.36|60.31|51.92|
> |QLSD-FT|93.80|89.30|91.56|87.80|79.49|65.35|37.44|27.74|
> |pFedGP|96.11|94.15|94.77|91.02|85.88|75.88|57.32|46.53|
> |**pFedVMP**|**97.23**|**95.60**|**96.97**|**95.09**|**88.12**|**80.81**|**64.22**|**56.75**|
>
> The table shows that pFedVMP achieves the highest test accuracy. This improvement stems from the ability of pFedVMP to guide local training through a regularization term based on global feature centroids, which reduces the risk of overfitting during local training. In contrast, FedPA, FedEP, QLSD, and pFedGP do not utilize global feature centroids to guide local model training.
>
> In the revised manuscript, we have cited these works.
>
> **W2/Q2 Computational cost of pFedVMP**: We agree with the reviewer that the computational cost of matrix inversion may be high. However, pFedVMP primarily aims to enhance model utility, which necessitates additional computation cost when leverage more information. We believe that performance improvement serves as the primary motivation for federated learning.
>
> Meanwhile, we note that some existing Federated learning methods that leverages the second-order statistics information such as covariance matrix or precision matrix, such as pFedGP and FedPAC, have similar computational costs. Specifically, let $Z$ denote the number of feature dimensions. pFedGP trains a personalized Gaussian process classifier on each clients, which requires to training a Gaussian kernel and has a computational complexity at $\mathcal{O}(Z\^3)$. FedPAC requires to optimize a quadratic function w.r.t. the covariance matrices of features, which also has a computational complexity at $\mathcal{O}(Z\^3)$. These methods have similar computational cost with pFedVMP.
>
> **W3/Q3 Hyperparameter Selection**: Thanks for this constructive comment. Equations 8 and 10 contains the following hyperparameters: the penalty scalar $\xi\_1$ and the scalar $\alpha$ that ensures the precision matrix $\boldsymbol{\Lambda}\_{k,n}\^\mathrm{z}$ is full rank. In the following, we discuss the selection values in the numerical experiments.
>
> - The penalty scalar $\xi\_1$: In the previous paper, we have investigated the effect of the penalty scalar $\xi\_1$ on the learning performance of pFedVMP. The results are presented in Appendix C.3.
> - The scalar $\alpha$ is defined to maintain that the precision matrix is full rank. Since the non-zero singular values of the pesudo inverse of the covariance matrix $(\boldsymbol{\Sigma}\_{k,n}\^\mathrm{z})\^\dagger$ is around $1e{3}$, we set the scalar $\alpha = 1$ in the numerical experiment.
>
> **W4/Q4 Theoretical guarantees**: As **Reviewer WGea's** suggestion, we have included additional derivations related to the variational message passing framework in Section 4, which enhance the theoretical foundation of the proposed method. However, we note that establishing a comprehensive convergence guarantee for the variational message passing framework remains an open problem in the research field. To date, there is a lack of rigorous convergence analyses for such methods. Consequently, even Bayesian federated learning approaches like FedEP and pFedGP do not provide detailed convergence analyses. Addressing this challenge is beyond the scope of this paper and is left for future investigation.

---

> > ### Comment · Reviewer_T9rb · 2024-12-01
> >
> > Thanks for the efforts in the responses. However, my primary concern for raising the rating is the lack of theoretical guarantees.

---

> > > ### Author Response · Authors · 2024-12-04
> > >
> > > **We thank Reviewer T9rb's comment.**
> > >
> > > We understand the concern regarding the lack of theoretical guarantees for the proposed method. Below, we provide additional clarification.
> > >
> > > 1. The guarantees of variational message passing are typically derived under certain constraints, such as marginalization or expectation constraints, to ensure the surrogate distribution closely approximates the target distribution. In our method, we employ expectation constraints, which align the mean and covariance of the surrogate distribution with those of the target distribution. The local updates in equations (10) and (11) and the aggregation in equation (13) are designed based on these expectation constraints.
> > > 2. Solving Problem (P2), a KL-divergence minimization problem, using variational message passing can be interpreted as an extended optimization problem that accounts for uncertainty. In the extreme case where uncertainty is entirely unknown, the KL-divergence minimization problem reduces to a conventional optimization problem. For example, FedPA [R2] addresses a federated least squares regression problem with a linear model, as presented in Equation (2). Additionally, Equation (3) demonstrates that the solution obtained by maximizing the likelihood distribution, i.e., the mean of the distribution, coincides with the solution derived from convex optimization.
> > > 3. According to prior literature [R1], rigorous convergence analyses of variational message passing are often based on state evolution, a theoretical tool used to track the dynamic behavior of variances during distribution updates. Since we estimate the variances (or covariances) of the distributions using sampling methods, the estimates may be biased due to the limited sample size. Consequently, existing works, such as FedPA [R2], rely on numerical experiments to evaluate the quality of the estimated gradient updates.
> > >
> > > Based on these discussions, we note that establishing theoretical guarantees is beyond the scope of this paper and is left for future exploration.
> > >
> > > [R1] Javanmard, Adel, and Andrea Montanari. "State evolution for general approximate message passing algorithms, with applications to spatial coupling." *Information and Inference: A Journal of the IMA* 2.2 (2013): 115-144.
> > >
> > > [R2] MaruanAl-Shedivat, Jennifer Gillenwater, Eric Xing, and Afshin Rostamizadeh. Federated learning via posterior averaging: A new perspective and practical algorithms. ICLR, 2021

---

### Official Review · Reviewer_5t2i · 2024-11-07

**Soundness:** 3
**Presentation:** 3
**Contribution:** 2
**Rating:** 5
**Confidence:** 3

**Summary:**

The paper proposes a personalized federated learning algorithm based on Bayesian estimation. The core idea is that clients learn a global shared model while they train a personalized head.

**Strengths:**

Personalization in federated learning by splitting the model into two parts such that one part is learned globally while the head is learned locally is proved to be effective. The contribution and motivation of the paper is clear.

**Weaknesses:**

The novelty of the paper is not high in my opinion. The core idea has been proposed before in the literature. One of the first work that I know which employ the same idea for personalized federated learning is FedRep of Collins et al., (2021). However, Collins et al., (2021) solves the problem using optimization. It seems that this paper solves the problem using Bayesian. Furthermore, Bayesian federated learning has been studied extensively before. Although, the paper compares the proposed algorithm against set of baselines, I think the paper misses the comparison with FedRep which is closely related to the study of this paper.

**Questions:**

I am not expert at Bayesian learning and I cannot evaluate the novelty of this work in this aspect. Can you explain the key differences between your proposed algorithm prior Bayesian federated learning? My concern is that if we can apply prior Bayesian federated learning to solve the problem which has been already studied by convex optimization techniques.

---

> ### Author Response · Authors · 2024-11-28
>
> **We thank Reviewer 5t2i for careful reading and constructive comments.**
>
> **W1.1 Novelty of our paper**
>
> `The novelty of the paper is not high in my opinion. The core idea has been proposed before in the literature.`
>
> We respectfully disagree with the comment. The central contribution of our work lies in leveraging the **mean and covariance** of global feature centroids to enhance the performance of personalized federated learning (pFL). This approach is novel compared to prior works on pFL, such as FedProto, GPFL, and FedPAC.
>
> Previous methods primarily estimate global feature centroids through arithmetic averaging of feature samples. However, due to the statistical heterogeneity of training data, the **arithmetic mean** may deviate from the true centroids. By utilizing variational message passing, pFedVMP provides more **precise estimates** of the distributions of global feature centroids, incorporating both their mean and covariance, thereby improving personalized model training performance.
>
> **W1.2 Comparison with FedRep**
>
> - **Similarity**: Both FedRep and pFedVMP consider to split the neural network model into a head and a base, where the base model aims to learn the common feature representations and the head model aims to achieve personalized goals.
> - **Difference**: FedRep does not involve a constraint of global feature centroids in local training. This increases the overfitting risk of the personalized model on the devices side. Unlike FedRep, pFedVMP is effective in boosting training accuracy and preventing overfitting by **regularizing** local training with global feature centroids. Meanwhile, benefiting from variational message passing, pFedVMP achieves more **precise estimates** of the distributions of global feature centroids, based on  the mean and the covariance of the global feature centroids, which enhancing the learning performance of personalized model training.
>
> **W1.3 & Q1 Key differences between pFedVMP and the prior Bayesian federated learning (BFL)**
>
> BFL can be broadly categorized into client-side BFL and server-side BFL in terms of FL architectures [1]. Client-side BFL focuses on learning Bayesian local models on client nodes, while server-side BFL aggregates local updates for global models using Bayesian methods. Our paper belongs to server-side BFL.
>
> The prior server-side BFL works related to our paper include FedPA, FedEP, QLSD, pFedBayes, which formulate model training as Bayesian inference tasks and aggregate the distributions of local **parameters**. FedPA (Al-Shedivat et al., 2021) approximated the posterior distribution into the product
> of distributions with respect to local datasets during local model training. FedEP (Guo et al., 2023)
> developed the Bayesian model aggregation rule by using expectation propagation. QLSD (Vono
> et al., 2022) extended the approach in FedPA with the quantized Langevin stochastic dynamics for
> local update. pFedBayes (Zhang et al., 2022) uses variational inference to approximate the posterior distribution of local model parameters for each client, and aggregates local models on the server.
>
> However, the above BFL methods do not utilize **global feature centroids** to guide local model training, which limits their ability to effectively address data heterogeneity. In contrast, pFedVMP considers both model **parameters** and **feature** centroids. The gain of pFedVMP comes from the following two folds: on one hand, pFedVMP guides the local training using a **regular** term of global feature centroids, decreasing the over-fitting risk of local training; on the other hand, pFedVMP achieves more **precise estimates** of the distributions of global feature centroid by variational message passing.
>
> [1] Bayesian Federated Learning: A Survey, IJCAI-23
>
> **Q2 Concern of the application of BFL**
>
> BFL is applicable to "the problem which has been already studied by convex optimization techniques".
>
> We interpret the phrase "the problem which has been already studied by convex optimization techniques" as referring to problems like Problem 3 in FedRep, which involves convex linear regression and is solvable via convex optimization methods. In such cases, the optimization problem of minimizing the loss function given some data samples is equivalent to estimating parameters using data samples by maximizing the likelihood or posterior distribution. For instance, FedPA (Al-Shedivat et al., 2021) addresses a federated least squares regression problem with a linear model, as shown in Equation (2). Furthermore, Equation (3) demonstrates that the solution obtained by maximizing the likelihood distribution, i.e., the mean of the distribution, aligns with the solution from convex optimization.

---

### Official Review · Reviewer_WGea · 2024-11-08

**Soundness:** 2
**Presentation:** 2
**Contribution:** 3
**Rating:** 6
**Confidence:** 3

**Summary:**

The paper presents a Bayesian approach to dealing with personalized federated learning (pFL). In particular, a "shared" base model is learned to map inputs to representations and each client learns a local head model to turn those representations into prediction outputs. In their approach, distributions of parameters of the base and head models are learned. In addition, the distribution of the representations are considered via a GMM (mixing on true labels). Locally, updates for the base, head, and GMM are learned. For global aggregation the base and GMM models are updated. The paper consider the case where the relevant distribution of the base and head models are Gaussians. Their approach is tested across various vision datasets.

**Strengths:**

- The approach shows promising experimental results, showing better results than the baselines reported
- The overall approach make intuitive sense, combining ideas from pFL and Bayesian FL.

**Weaknesses:**

- The primary weakness of the paper is in part of its presentation. This is particularly the case for Section 4 where conceptual optimization goals is mixed in with practical simplifications.
 - In addition, there are no detailed derivation for some of the quantities used (in main text nor appendix).

**Questions:**

Questions + Remarks:

1. $p({\theta}^{\\rm b}, \\{ \theta_n^{\\rm h} \\}, \\{ z_k \\}, S)$ is a distribution over parameters (+ representations) and samples $S$. But, as far as I can tell from the equations, the surrogate distribution $q$ being consider is only over parameters (+ representations). As such, it is unclear how the KL-divergences are being evaluated, eg, (P1) including the argmin.
Please provide clarity on this support issue of the distributions and how the KL-divergence is being evaluated.

2. From what I understand, when (P1) is referred in text, it only refers to the parameter updates and not the variational / KL-divergence aspect over the equation. This is rather unclear in Section 4.2.
Please clarify this (P1), perhaps by presenting the entire optimization in multiple line (labeling as (P1a) and (P1b) for instance).

3.  I think additional clarity in the text should also be added to distinguish section which are considering the update of parameters in (P1) (Section 4.3) vs updates on the distribution in (P2) (Section 4.2 & 4.3). This aspect is also mixed in Section 4.3.1. It may be worth splitting this subsubsection into two separate subsubsections, one for local updates on the parameters and one for local updates on the distributions.

4. Is (P2) and (P3) equivalent (when restricting optimization to local parameter etc)? Line 269-270 says that the optimization is converted. Does this imply equivalence?

5. The soundness of going from (P3) to (8) is a bit unclear. Could you please elaborate on the derivation (which I believe is just maximizing (7)) and why it is "low-cost implementation of SG-MCMC".

6. (P2) seems to be imprecise. In particular, the second term in the KL is not a normalized distribution? In particular, several "prior" distribution seem to be missing in (P2) and the accompanying text. Please clarify this.
It would also be useful for completeness to include a derivation of the specific factorization you are using; and its subsequent use in (P3).

---

Minor:
 - References were cut off from the main text.
 - The subscript of the max in (1) and (P1) is not very nice. Maybe having the subscript fully under the "max" would improve readability of these equations.
 - $\\mathcal{Z}_{k,n}$ seems to be incorrect on line 160 (should have $k$ instead of $y_k$)
 - I think \bigcup $\\bigcup$ is typically used over \cup $\\cup$ for indexed unions.
 - (1) and 2) on line 42 - 44 are not consistent
 - Line 240 in denominator, there is a missing bracket
 - (P1-3) should be on the RHS to be consistent with equation numbering (maybe via \tag)
 - Figure 3, missing space after "Upper:"

---

> ### Author Response · Authors · 2024-11-28
>
> **We thank Reviewer WGea for careful reading and constructive comments.**
>
> **W1** Thanks for the comment. As the suggestions from the reviewer, we have improved the presentation of the paper, especially Section 4.
>
> **W2** Thanks for the comment. In the revised manuscript, we provide clearer explanations and derivations to make the quantities used, improving the readability of this paper.
>
> **Q1**: Thanks for pointing out this issue. We agree with the reviewer that the surrogate distribution $q(\boldsymbol{\theta}\^{\mathrm{b}}, \{\boldsymbol{\theta}\_n\^{\mathrm{h}}\}, \{\mathbf{z}\_k\})$ is defined over the parameters (and representations) $(\boldsymbol{\theta}\^{\mathrm{b}}, \{\boldsymbol{\theta}\_n\^{\mathrm{h}}\}, \{\mathbf{z}\_k\})$, while $p(\boldsymbol{\theta}\^{\mathrm{b}}, \{\boldsymbol{\theta}\_n\^{\mathrm{h}}\}, \{\mathbf{z}\_k\}, \mathcal{S})$ includes the sample set $\mathcal{S}$. This difference indeed raises questions about the proper evaluation of the KL-divergence.
>
> To address this issue, we use the posterior distribution $p(\boldsymbol{\theta}\^{\mathrm{b}}, \{\boldsymbol{\theta}\_n\^{\mathrm{h}}\}, \{\mathbf{z}\_k\} | \mathcal{S})$ to replace the joint distribution  $p(\boldsymbol{\theta}\^{\mathrm{b}}, \{\boldsymbol{\theta}\_n\^{\mathrm{h}}\}, \{\mathbf{z}\_k\}, \mathcal{S})$ in (P1). This change ensures that the distributions $p$ and $q$ are defined on the same support (parameters and representations), allowing for a consistent evaluation of the KL-divergence.
>
> **Q2**: Thanks for the comment. We agree with the reviewer that the presentation of (P1) may cause confusion for readers. In our approach, we use a surrogate distribution $q(\boldsymbol{\theta}\^{\mathrm{b}}, \{\boldsymbol{\theta}\_n\^{\mathrm{h}}\}, \{\mathbf{z}\_k\})$ to approximate the distribution $p$. Consequently, estimating the parameters and feature centroids $(\boldsymbol{\theta}\^{\mathrm{b}}, \{\boldsymbol{\theta}\_n\^{\mathrm{h}}\}, \{\mathbf{z}\_k\})$ involves maximizing the surrogate distribution $q$. Since each factor of distribution $q$ is either Gaussian distribution or GM distribution, the solution of the maximization is the mean of each Gaussian distribution (or each Gaussian component of the GM distribution). Thus, our primary focus is on updating the distribution to minimize the KL divergence between $p$ and $q$. The revised (P1) is provided as follows:
> $$
> \begin{align}
> \mathrm{(P1)} \min\_{q(\boldsymbol{\theta}\^{\mathrm{b}}, \{\boldsymbol{\theta}\_n\^{\mathrm{h}}\}, \{\mathbf{z}\_k\})} D\_\mathrm{KL} \left(p(\boldsymbol{\theta}\^{\mathrm{b}}, \{\boldsymbol{\theta}\_n\^{\mathrm{h}}\}, \{\mathbf{z}\_k\} | \mathcal{S}) \\| q(\boldsymbol{\theta}\^{\mathrm{b}}, \{\boldsymbol{\theta}\_n\^{\mathrm{h}}\}, \{\mathbf{z}\_k\}) \right).
> \end{align}
> $$
> **Q3**: Thanks for the comment. Since each factor of distribution $q$ is either Gaussian distribution or GM distribution, the MAP estimate is taking the mean of each Gaussian distribution (or each Gaussian component of the GM distribution). To make this clear, we added extra clarity in Section 4.3.2.
>
> **Q4**: (P3) is not equivalent to (P2), but is an approximation to (P2) in the FL setting. Specifically, consider the objective function of (P2), given by
> $$
> \begin{equation*}
>     \text{(P2)} \min\_{q\_n(\boldsymbol{\theta}\^{\mathrm{b}}, \boldsymbol{\theta}\_n\^{\mathrm{h}}, \{\mathbf{z}\_k\})}
>     D\_\mathrm{KL}
>     \left(p(\boldsymbol{\theta}\^{\mathrm{b}}, \{\boldsymbol{\theta}\_n\^{\mathrm{h}}\}, \{\mathbf{z}\_k\} | \mathcal{S})\\|
>     q\_n(\boldsymbol{\theta}\^{\mathrm{b}}, \boldsymbol{\theta}\_n\^{\mathrm{h}}, \{\mathbf{z}\_k\}) q\_{-n}(\boldsymbol{\theta}\^{\mathrm{b}}) q\_{-n}(\{\mathbf{z}\_k\})
>     q\_{-n}(\{\boldsymbol{\theta}\_n\^{\mathrm{h}}\}) \right).
> \end{equation*}
> $$
> In the FL setting, each client can only assess the local dataset $\mathcal{S}\_n$, or a subset of training data sample set $\mathcal{S}$. This limitation makes it difficult to update $q\_n(\boldsymbol{\theta}\^{\mathrm{b}}), q(\boldsymbol{\theta}\_n\^{\mathrm{h}})$ by drawing samples of $\boldsymbol{\theta}\^{\mathrm{b}}, \boldsymbol{\theta}\_n\^{\mathrm{h}}$ from the joint distribution $p$ directly. To address this, we define the surrogate distribution $\tilde{q}\_{n}$ to approximate the joint distribution $p$ by fixing the cavity factors $q\_{-n}(\boldsymbol{\theta}\^{\mathrm{b}})$, $q\_{-n}(\{\mathbf{z}\_k\})$, $q\_{-n}(\{\boldsymbol{\theta}\_n\^{\mathrm{h}}\})$ on the side of client $n$, given by
> $$
> \begin{align*}
>     &\tilde{q}\_n(\boldsymbol{\theta }\^{\mathrm{b}},\boldsymbol{\theta }\_n\^{\mathrm{h}},\{\mathbf{z}\_{k}\})
>     =p(\mathcal{S}\_n | \boldsymbol{\theta }\^{\mathrm{b}},\boldsymbol{\theta }\_n\^{\mathrm{h}}) q\_n(\{\mathbf{z}\_{k}\})
>     q\_{-n}(\boldsymbol{\theta}\^{\mathrm{b}}) q\_{-n}(\{\mathbf{z}\_k\})
>     q\_{-n}(\{\boldsymbol{\theta}\_n\^{\mathrm{h}}\}) ).
> \end{align*}
> $$
> Based on $\tilde{q}\_{n}$, we obtain the local optimization problem on client $n$ in Problem (P3).

---

> > ### Author Response · Authors · 2024-11-28
> >
> > **Q5.1** **The derivation from (P3) to (8):**  We agree with the reviewer that this derivation is based on maximizing (7). We now derive the loss function of SGD (8). Based on the above definition of $\tilde{q}\_n(\boldsymbol{\theta }\^{\mathrm{b}},\boldsymbol{\theta }\_n\^{\mathrm{h}},\{\mathbf{z}\_{k}\})$, the negative logarithm of the target distribution is expressed as:
> > $$
> > \begin{align*}
> >     -\log \tilde{q}\_n(\boldsymbol{\theta }\^{\mathrm{b}},\boldsymbol{\theta }\_n\^{\mathrm{h}},\{\mathbf{z}\_{k}\})
> >     &=-\log p(\mathcal{S}\_n | \boldsymbol{\theta }\^{\mathrm{b}},\boldsymbol{\theta }\_n\^{\mathrm{h}})- \log q(\{\mathbf{z}\_k\})
> >     - \log q\_{-n}(\boldsymbol{\theta}\^{\mathrm{b}})
> >     - \log q\_{-n}(\{\boldsymbol{\theta}\_n\^{\mathrm{h}}\}) + \mathrm{Const.}
> > \end{align*}
> > $$
> > On client $n$, computing the cavity factors $q\_{-n}(\boldsymbol{\theta}\^{\mathrm{b}})$ and $q\_{-n}(\{\boldsymbol{\theta}\_n\^{\mathrm{h}}\})$ may lead to instability during sampling. Thus, we exclude the terms involving $q\_{-n}(\boldsymbol{\theta}\^{\mathrm{b}}), q\_{-n}(\{\boldsymbol{\theta}\_n\^{\mathrm{h}}\})$, resulting in the following simplified loss function:
> > $$
> > \begin{align*}
> >     -\log p(\mathcal{S}\_n | \boldsymbol{\theta }\^{\mathrm{b}},\boldsymbol{\theta }\_n\^{\mathrm{h}})- \log q(\{\mathbf{z}\_k\})
> > \end{align*}
> > $$
> > By assuming the data samples are i.i.d., we obtain eq. (8):
> > $$
> > \begin{align}
> >     \sum\nolimits\_{i = 1}\^{S\_n}
> >      \left( - \log p( \mathbf{x}\_{n,i},y\_{n,i}|\boldsymbol{\theta }\^{\mathrm{b}},\boldsymbol{\theta }\_n\^{\mathrm{h}})+\xi\_1  \\|\mathbf{z}\_{n,i}-\boldsymbol{\mu}\_{y\_{n,i}}\^{\mathrm{z}}\\|\^2 \right), \tag{8}
> > \end{align}
> > $$
> > where the second term is because calculating the precision matrix $\boldsymbol{\Lambda}\_{y\_{n,i}}\^{\mathrm{z}}$ in the loss function may cause the gradient unstable, and we use a spherical Gaussian distribution with the mean $\boldsymbol{\mu}\_{y\_{n,i}}\^{\mathrm{z}}$ and the precision matrix $\xi\_1 \mathbf{I}$ instead.
> >
> > **Q5.2** **"A low-cost implementation of SG-MCMC"**: The results of $\boldsymbol{\theta}\^{\mathrm{b}}, \boldsymbol{\theta}\_n\^{\mathrm{h}}$ updated by SGD can be regarded as a single sample drawn by SG-MCMC, which reduces the computational and storage cost in the sampling.
> >
> > **Q6** We agree with the reviewer that in the previous version, the second term of the KL-divergence in (P2) is incorrect. The renewed (P2) is given by
> > $$
> > \begin{equation*}
> >     \text{(P2)} \min\_{q\_n(\boldsymbol{\theta}\^{\mathrm{b}}, \boldsymbol{\theta}\_n\^{\mathrm{h}}, \{\mathbf{z}\_k\})}D\_\mathrm{KL}(p(\boldsymbol{\theta}\^{\mathrm{b}}, \{\boldsymbol{\theta}\_n\^{\mathrm{h}}\}, \{\mathbf{z}\_k\} | \mathcal{S}) \\| q\_n(\boldsymbol{\theta}\^{\mathrm{b}}, \boldsymbol{\theta}\_n\^{\mathrm{h}}, \{\mathbf{z}\_k\}) q\_{-n}(\boldsymbol{\theta}\^{\mathrm{b}}) q\_{-n}(\{\mathbf{z}\_k\})
> >     q\_{-n}(\{\boldsymbol{\theta}\_n\^{\mathrm{h}}\}) ).
> > \end{equation*}
> > $$
> > The derivation is given as following. We expand $q(\boldsymbol{\theta}\^{\mathrm{b}}, \{\boldsymbol{\theta}\_n\^{\mathrm{h}}\}, \{\mathbf{z}\_k\})$, i.e., the second term of KL-divergence as
> > $$
> > \begin{align*}
> >     q(\boldsymbol{\theta}\^{\mathrm{b}}, \{\boldsymbol{\theta}\_n\^{\mathrm{h}}\}, \{\mathbf{z}\_k\})
> >     & \overset{(a)}{\propto}
> >     q(\boldsymbol{\theta}\^{\mathrm{b}})
> >     q(\{\boldsymbol{\theta}\_n\^{\mathrm{h}}\})
> >     q(\{\mathbf{z}\_k\})  \\\\
> >     & \overset{(b)}{\propto} \left( q\_\mathrm{pri}(\boldsymbol{\theta}\^{\mathrm{b}})\prod\_{n=1}\^N q\_n(\boldsymbol{\theta}\^{\mathrm{b}}) \right)
> >     \left(\prod\_{n=1}\^Nq\_\mathrm{pri}(\boldsymbol{\theta}\_n\^{\mathrm{h}})
> >     q\_n(\boldsymbol{\theta}\_n\^{\mathrm{h}})\right)
> >     \left(q\_\mathrm{pri}(\{\mathbf{z}\_k\})\prod\_{n=1}\^N q\_n(\{\mathbf{z}\_k\}) \right) \\\\
> >     & \overset{(c)}{\propto} q\_n(\boldsymbol{\theta}\^{\mathrm{b}}) q\_n(\boldsymbol{\theta}\_n\^{\mathrm{h}}) q\_n(\{\mathbf{z}\_k\}) q\_{-n}(\boldsymbol{\theta}\^{\mathrm{b}}) q\_{-n}(\{\mathbf{z}\_k\})  q\_{-n}(\{\boldsymbol{\theta}\_n\^{\mathrm{h}}\})
> > \end{align*}
> > $$
> >
> > where step (a) is because of the definition of $q(\boldsymbol{\theta}\^{\mathrm{b}}, \{\boldsymbol{\theta}\_n\^{\mathrm{h}}\}, \{\mathbf{z}\_k\})$ in eq. (3),  step (b) is obtained by plugging eq. (4) into the definition of $q(\boldsymbol{\theta}\^{\mathrm{b}}, \{\boldsymbol{\theta}\_n\^{\mathrm{h}}\}, \{\mathbf{z}\_k\})$, and step (c) is due to the definition of the cavity factors $q\_{-n}(\boldsymbol{\theta}\^{\mathrm{b}}), q\_{-n}(\{\mathbf{z}\_k\}),  q\_{-n}(\{\boldsymbol{\theta}\_n\^{\mathrm{h}}\})$.

---

> > > ### Author Response · Authors · 2024-11-28
> > >
> > > ***Minors***: We thank the reviewer for the attention to detail. We have thoroughly reviewed the paper and addressed all similar issues. The following corrections have been implemented:
> > >
> > > 1. We have uploaded the completed version of the paper, including references and appendices.
> > > 2. We agree with the reviewer and the subscript is fully moved under the "max" operator in (1) and (P1).
> > > 3. We have fixed this error.
> > > 4. We agree with the reviewer and use \bigcup in the revised manuscript.
> > > 5. We have fixed the inconsistent numbering.
> > > 6. The missing brackets have been added.
> > > 7. Equation numbering has been included for all problems.
> > > 8. The typo has been corrected.

---

> > > > ### Comment · Reviewer_WGea · 2024-11-29
> > > >
> > > > Thank you for the detailed responses and manuscript changes.
> > > > The responses have adequately answered my questions and concerns, and I will raise my score accordingly.
> > > >
> > > > Some other minor issues I found whilst reading the relooking at the manuscript:
> > > >  - (1) is used for "listing items" in-text and also for equations. One may want to change this to avoid confusion.
> > > >  - I would still recommend for the an avoidance of double equation numbering, eg, (P3) and (8a) + (8b). I think it would be better to have (P3a) and (P3b) only and have (P3) when referencing both. Of course, this a matter of taste.
> > > >  - One may want to be mindful that some of the equation with multiple definitions on a single line are quite busy, eg, (4), (6), and (11). Some whitespace between each equation could help with readability.

---

> > > > > ### Author Response · Authors · 2024-11-30
> > > > >
> > > > > **We sincerely thank Reviewer WGea for this positive feedback.**
> > > > >
> > > > > We will further revise the manuscript to address these minor issues, including clarifying item and equation numbering, avoiding double equation numbering where possible, and improving the readability of equations by adding appropriate whitespace.
> > > > > These revisions will be included in the camera-ready version if the manuscript is accepted.

---

### Meta-Review · Area_Chair_BzHs · 2024-12-18

**Metareview:**

A paper on an interesting topic, which unfortunately does not pass the bar for acceptance. In revising their manuscript, I strongly encourage the authors to take into account the comments of the reviewers, in particular those of T9rb that the authors have not thoroughly discussed, especially the question of formal guarantees. Such guarantees would be important to better sell the ideas of the paper, and I disagree on the authors' claim that they are out of scope of the paper.

**Additional Comments On Reviewer Discussion:**

Though the authors' feedback came relatively late during the review process, they did a decent job at it, but failed to properly comment on comments on formal guarantees of their method (T9rb).

---

### Decision · Program_Chairs · 2025-01-22

Reject